# SuperHype: Hypergraph Generation via Graph-Superposition Decomposition

Lucas Gantès [1 2]   Abele Mălan [3]   Roberto Gheda [4]   Robert Birke [5]   Lydia Chen [3 4]

## Abstract

Hypergraphs are graph generalizations with key applications in domains such as healthcare, where strict data privacy requirements apply, or bioinformatics, where testing new compounds is costly. However, due to their combinatorial nature, hypergraph representations are often either intractable or lead to significant information loss. For this reason, research into hypergraph synthesis is limited, and state-of-the-art approaches yield poor generation quality in terms of overall structural patterns and graph-level validity. To address such shortcomings, we introduce SuperHype, an exact and tractable hypergraph diffusion model. The core of SuperHype is the graph-superposition decomposition, a novel representation that embeds a hypergraph into a multi-layer graph, enabling a tractable representation with no loss of generalization. To generate new samples from such representations, we introduce a Graph-Superposition Transformer that treats the superposition as an interconnected sequence of layers. Moreover, we enhance the model's performance by incorporating hypergraph-specific auxiliary features and aggregating indirect node interactions via triplet pooling. Our evaluation across five datasets shows that SuperHype generally reproduces local and global connectivity patterns with superior fidelity compared to state-of-the-art baselines.

## 1. Introduction

Hypergraphs model intricate, high-order relationships across diverse domains, including social networks (Enduri, 2024), bioinformatics (Feng et al., 2021), or recommender systems (Xia et al., 2022), and are also used in tasks such

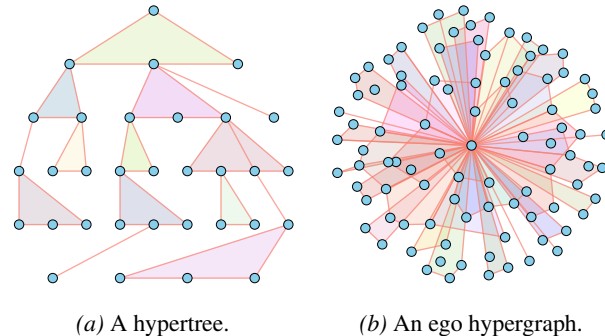

*(a)* A hypertree.   *(b)* An ego hypergraph.

*Figure 1.* Hypergraphs are a generalization of graphs in which *hyperedges* connect an arbitrary number of nodes.

as node classification and link prediction (Gao et al., 2026; Bazaga et al., 2024). Figure 1 showcases two examples of hypergraphs. To train such models on sensitive inputs, synthetic data generated by generative models has emerged as a prominent solution across various data types, such as images (Suh et al., 2025), tables (Kotelnikov et al., 2023), and hypergraphs (Wang et al., 2018). However, state-of-the-art hypergraph generators such as HyperPLR (Wen & Yu, 2025) and HYGENE (Gailhard et al., 2025) remain limited, as they rely on representations or generation procedures that are not fully suited to the higher-order structure of hypergraphs.

The main challenge in synthesizing hypergraphs lies in their complex representation. While traditional graphs with $|\mathcal{V}|$ nodes can be efficiently represented in $\mathcal{O}(|\mathcal{V}|^2)$ via adjacency matrices, the number of possible hyperedges in a hypergraph is $\mathcal{O}(2^{|\mathcal{V}|})$. A common representation paradigm is the *bipartite representation* (Figure 2b), which embeds a hypergraph with $|\mathcal{V}|$ nodes and $|\mathcal{E}|$ edges as a bipartite graph with $|\mathcal{V}| + |\mathcal{E}|$ nodes, in which original nodes are connected to the hyperedges containing them. Other representation paradigms, such as the *clique expansion* (Figure 2c), are tractable but introduce ambiguity in their embeddings. Hence, one representation can belong to multiple hypergraphs. Figure 2 showcases these problems in detail.

To enable the generation of synthetic hypergraphs without sacrificing the rich structural information they contain, we propose SuperHype, the first *diffusion model for hypergraph generation*. SuperHype makes hypergraph synthesis tractable via a **graph-superposition decomposition**,

---

[1]École Polytechnique, France [2]Télécom Paris, France [3]University of Neuchâtel, Switzerland [4]Delft University of Technology, Netherlands [5]University of Turin, Italy. Correspondence to: Lucas Gantès <lucas.gantes@polytechnique.org>, Abele Mălan <abele.malan@unine.ch>.

*Proceedings of the 43rd International Conference on Machine Learning*, Seoul, South Korea. PMLR 306, 2026. Copyright 2026 by the author(s).

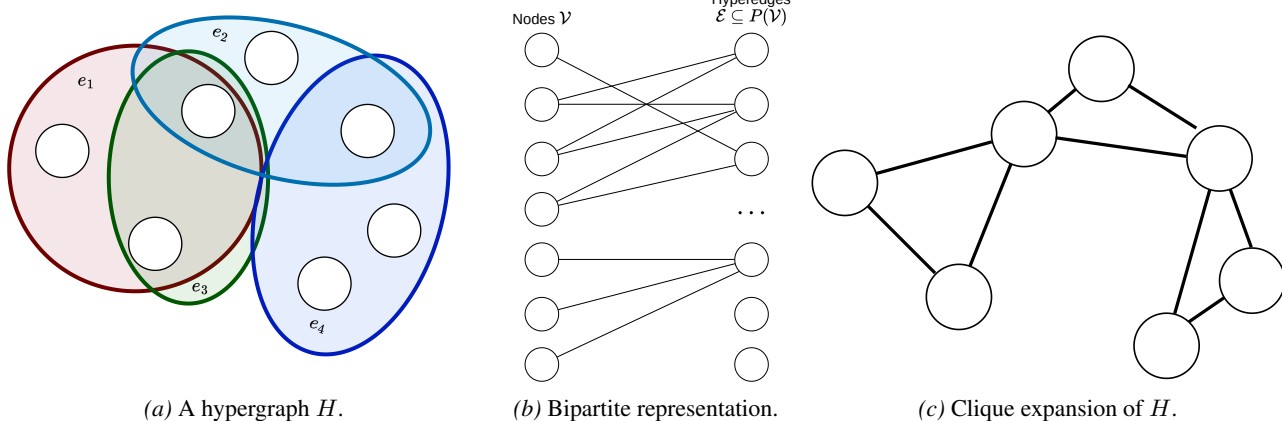

*(a)* A hypergraph $H$.  ·  *(b)* Bipartite representation.  ·  *(c)* Clique expansion of $H$.

*Figure 2.* Hypergraph representation is an NP-complete problem. (a) A hypergraph $H$; (b) The bipartite representation requires an extensive enumeration of the $\mathcal{O}(2^{|\mathcal{V}|})$ possible edges; (c) Using the clique expansion, the presence of the edge $e_3$ is lost due to ambiguity.

a novel algorithm that embeds a hypergraph into a multi-layer graph, thereby introducing tractability while retaining unambiguous representation. Unlike classical clique expansion, the mapping from a graph-superposition to a hypergraph is injective, meaning that the projection into a graph-superposition does not induce a loss of information. To handle these representations in the context of diffusion models, we propose a **Graph-Superposition Transformer**, a transformer-based diffusion model that performs message passing both within individual graph layers and across layers in the superposition. Coupled with a discrete denoising-diffusion process, the model produces high-fidelity synthetic hypergraphs that faithfully reproduce real-world patterns. Finally, for sharper control over maximal-clique formation, we augment the model with a **triplet aggregation mechanism** and specific auxiliary features. Together, these additions markedly improve the accuracy of the generated hypergraphs' local motifs and global topology.

We evaluate SuperHype's generative performance against six state-of-the-art baselines on five different datasets and demonstrate that SuperHype consistently achieves superior synthesis quality. Our contributions are as follows:

- SuperHype introduces the graph-superposition, a compact and exact representation for hypergraphs, along with a projection algorithm to construct such superpositions from hypergraphs.

- SuperHype introduces the Graph-Superposition Transformer, a novel neural network architecture for graph diffusion that enables high-fidelity hypergraph synthesis with tractable cost.

- SuperHype includes a triplet-aggregation mechanism that enables improved maximal clique formation abilities, resulting in superior synthesis quality.

- Our experiments across five datasets demonstrate that SuperHype consistently gives superior generation quality compared to six state-of-the-art baseline generators.

Our code is available online.

## 2. Related Work

**Graph generative models** Graph generation has been at the forefront of scientific research, with applications ranging from molecular design to social network simulation. Early deep generative approaches – based on variational autoencoders (Simonovsky & Komodakis, 2018), GANs (Wang et al., 2018), or autoregressive schemes (You et al., 2018) – struggled with graph-specific constraints like permutation equivariance, discreteness, and sparsity. More recent diffusion-based formulations better address these challenges (Vignac et al., 2023; Jo et al., 2024).

**Diffusion models** have emerged as powerful generative frameworks across continuous and discrete data modalities. *Denoising Diffusion Probabilistic Model* (DDPM) (Ho et al., 2020) defines a Markov chain that iteratively injects noise into the data, i.e., the forward process. A neural network is trained to reverse this process, enabling generation from a simple, tractable distribution such as Gaussian noise.

For applying diffusion models to graphs, both discrete and continuous approaches have been proposed. On the one hand, *DiGress* (Vignac et al., 2023) introduces a denoising diffusion process that preserves the discrete, sparse nature of adjacency matrices and proposes a graph transformer with auxiliary graph properties. On the other hand, *GruM* (Jo et al., 2024) models global topology as a mixture of diffusion processes, while *Cometh* (Siraudin et al., 2025) allows flexible control of the denoising schedule to trade off synthesis quality and computation. These advances yield markedly

improved graph connectivity and realism, though often at higher computational cost. Overall, while standard graph diffusion models can provide a solid base for the hypergraph setting, existing models are ill-suited to generating richer structures due to their explicit reliance on simple two-node edges and, often, on the adjacency matrix format.

Orthogonal work also exists on improving the scalability of graph generation through compact or hierarchical representations, such as K2-tree-based graph generation and hierarchical graph generative networks (Jang et al., 2024; Karami, 2024), or through more efficient discrete diffusion formulations (Qin et al., 2025b; Chen et al., 2023). Such directions complement our work: they cover techniques for reducing generation cost in standard graphs, whereas SuperHype focuses on an exact discrete representation for hypergraph generation.

**Hypergraph generative models** Hypergraphs generalize graphs by allowing *hyperedges* to connect more than two nodes, enabling richer modeling of higher-order relationships. Classical approaches infer hypergraphs from graphs, for example, through Bayesian reconstruction (Young et al., 2021) or machine learning methods trained to invert clique expansion (Wang & Kleinberg, 2024). GAN-based approaches (Pan et al., 2021) and variational autoencoders (Su et al., 2025) offer initial solutions, while HyperPLR (Wen & Yu, 2025) projects hypergraphs into weighted clique graphs before reconstruction via greedy algorithms. Nevertheless, HyperPLR generates a clique expansion of a hypergraph and then tries to convert it into a hypergraph using a greedy algorithm. This transformation induces a loss of information in the generated hypergraph. More recently, HYGENE (Gailhard et al., 2025) directly applies denoising diffusion to hypergraph generation, using iterative expansion. However, because expansion generates larger hypergraphs through repeated local extensions rather than modeling the full hypergraph structure jointly, errors can accumulate across steps, leading to suboptimal generation quality. Overall, the current state of the art falls short at capturing specific hypergraph properties, resulting in limited synthesis quality.

Recent latent-space approaches address complementary hypergraph generation settings. For example, ReLaSH (Ma et al., 2026) reconstructs joint latent spaces for hypergraphs with hyperlink attributes, where latent diffusion is useful for jointly modeling structure and attributes. In contrast, our setting focuses on the structural topology of unattributed hypergraphs. We therefore operate directly in a discrete graph-superposition state space, following evidence from discrete graph diffusion that data-space diffusion better preserves complex graph-level validity properties (Vignac et al., 2023; Qin et al., 2025a).

## 3. SuperHype

We start by going over preliminaries and notation for our hypergraph generation problem in Section 3.1. Then, we introduce our first main contribution in Section 3.2: *graph-superposition decomposition* for hypergraphs. To enable generation over this newly introduced graph-superposition representation, which takes the form of a small collection of related graphs, we extend discrete graph diffusion models into the *Graph-Superposition Transformer* as described in Section 3.3. Finally, to further improve the performance of SuperHype, we augment our model with *hypergraph-specific auxiliary features* and *triplet aggregation*, enhancing its ability to model higher-order graph properties and interactions between nodes with a common neighbor.

### 3.1. Preliminaries

While the set of possible edges in a graph $G = (\mathcal{V}, E)$ is $\mathcal{O}(|\mathcal{V}|^2)$, the set of all possible hyperedges in a hypergraph $H = (\mathcal{V}, \mathcal{E})$, with nodes $\mathcal{V} = \{1, 2, \dots\}$ and hyperedges $\mathcal{E} \subseteq \mathcal{P}(\mathcal{V})$, is $\mathcal{O}(2^{|\mathcal{V}|})$, where $\mathcal{P}$ denotes the power set. Hence, representing hypergraphs is NP-complete. A naive way to represent hypergraphs is as a *bipartite graph* with a set of nodes $\mathcal{V} \cup \mathcal{E}$, where the original nodes are connected to the hyperedges they belong to. This representation is extensive but introduces prohibitive scaling costs of $\mathcal{O}(2^{|\mathcal{V}|})$.

**Clique expansion** To mitigate the prohibitive cost of hypergraph representation, a mechanism based on *clique expansion* is commonly used.

**Definition 3.1** (Clique expansion of a hypergraph)**.** Let $H = (\mathcal{V}, \mathcal{E})$ be a hypergraph. Its *clique expansion* is the graph $G = (\mathcal{V}', \mathcal{E}')$ satisfying the following conditions:

  i. **Same set of vertices**: $\mathcal{V}' = \mathcal{V}$.

  ii. **Adjacency via hyperedge**: $(\{i, j\} \in \mathcal{E}' \iff \exists e \in \mathcal{E}, \{i, j\} \subseteq e), \forall i, j \in \mathcal{V}^2$.

The clique expansion of a hypergraph defined in Definition 3.1 transforms the higher-order relationships of hyperedges into pairwise connections. Consequently, it induces a loss of information in the generated hypergraph. Indeed, a clique-expansion representation may correspond to multiple hypergraphs, because every subset of a clique remains a clique. In contrast, a subset of a hyperedge does not necessarily form a hyperedge. For example, if a hypergraph contains two overlapping or nested hyperedges, their clique expansions may be indistinguishable from that of a single larger hyperedge. In contrast, our superposition preserves these distinctions by placing hyperedges that would otherwise create ambiguous maximal cliques in different layers.

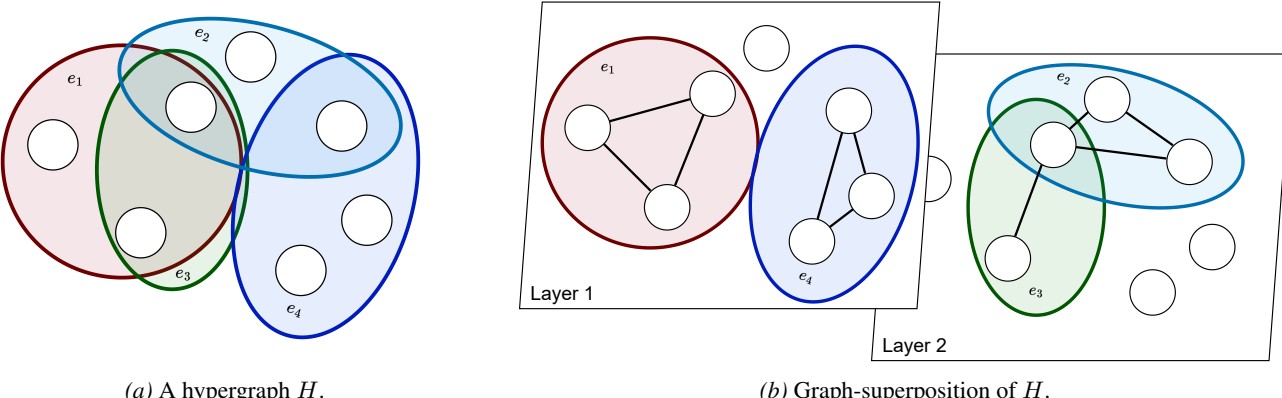

*(a)* A hypergraph $H$.         *(b)* Graph-superposition of $H$.

*Figure 3.* The proposed graph-superposition is a tractable representation that retains unambiguity. (a) A hypergraph $H$ with 7 nodes and 4 hyperedges; (b) A graph-superposition of $H$ on 2 layers: retains exactness while being tractable.

---

**Algorithm 1** Graph superposition projection

**Input:** Hypergraph $H = (\mathcal{V}, \mathcal{E})$,
enumeration $(e_i)_{i=1}^{|\mathcal{E}|}$ of elements in $\mathcal{E}$,
number of layers $L$
**Output:** Graph-superposition $(\mathcal{V}, (\mathcal{E}_l)_{l=1}^{L})$
1: $(\mathcal{E}_l)_{l=1}^{L} \leftarrow (\emptyset)_{l=1}^{L}$
2: **for** $i \leftarrow 1$ **to** $|\mathcal{E}|$ **do**
3:      $\pi \leftarrow \texttt{Rand\_Perm}(1, \ldots, L)$
4:      **for** $l' \leftarrow 1$ **to** $L$ **do**
5:          $l \leftarrow \pi_{l'}$
6:          $\mathcal{E}' \leftarrow \mathcal{E}_l \cup \mathcal{P}_2(e_i)$
7:          $\mathcal{C} \leftarrow \texttt{Maximal\_Clique}(\mathcal{V}, \mathcal{E}_l)$    $\{\mathcal{O}(3^{|\mathcal{V}|/3})\}$
8:          $\mathcal{C}' \leftarrow \texttt{Maximal\_Clique}(\mathcal{V}, \mathcal{E}')$    $\{\mathcal{O}(3^{|\mathcal{V}|/3})\}$
9:          **if** $e_i \notin \mathcal{C} \wedge \mathcal{C}' = \mathcal{C} \cup \{e_i\}$ **then**
10:             $\mathcal{E}_l \leftarrow \mathcal{E}'$
11:             Break
12:          **else**
13:             **return** Fail
14:          **end if**
15:      **end for**
16: **end for**

---

### 3.2. Superposition transform

The intractability of hypergraph representation prevents its application to diffusion models, whose computational and memory costs scale with the size of the data. Hence, current methodologies for hypergraph synthesis limit these costs by either relying on graph extrapolation or ambiguous hypergraph representations. However, this unfortunately degrades the quality of these models' generation.

To overcome this limitation, we develop a **graph-superposition decomposition**, which enables tractable and unambiguous hypergraph representation. By decomposing the hypergraph into multiple layered subgraphs, SuperHype manages to reduce the computational cost of hypergraph

representation from $\mathcal{O}(|\mathcal{V}||\mathcal{E}|) = \mathcal{O}(|\mathcal{V}|2^{|\mathcal{V}|})$ to $\mathcal{O}(|\mathcal{V}|^2 L)$, where $L$ is the number of layers used to store the graph. Here, the challenge is twofold: minimizing *memory cost* and guaranteeing *unambiguity*. First, since the number of total edges is $\mathcal{O}(2^{|\mathcal{V}|})$, they must be spread over the layers as much as possible to contain the memory costs. Second, the hyperedges must be distributed to avoid ambiguity and enable exact reconstruction. To overcome these challenges, we propose a greedy algorithm that maps a hypergraph $H = (\mathcal{V}, \mathcal{E})$ to a *graph-superposition representation* $(\mathcal{V}, (\mathcal{E}_l)_{l=1}^{L})$ in $\mathcal{O}(3^{|\mathcal{V}|/3} L |\mathcal{E}|)$. We formalize the graph-superposition of a hypergraph in Definition 3.3.

**Definition 3.2** (Graph-superposition). Let $L \in \mathbb{N}^*$. A *graph-superposition* $(\mathcal{V}, (\mathcal{E}_l)_{l=1}^{L})$ with $L$ layers is a $L$-tuple of graphs, each with nodes $\mathcal{V}$. For each $l \in [\![1, L]\!]$, $(\mathcal{V}, \mathcal{E}_l)$ is a graph.

**Definition 3.3** (Graph-superposition representation of a hypergraph). Let $H = (\mathcal{V}, \mathcal{E})$ be a hypergraph. A *graph-superposition representation* of $H$ is a graph-superposition $(\mathcal{V}, (\mathcal{E}_l)_{l=1}^{L})$ sharing the same node set $\mathcal{V}$ as $H$ so that the hyperedge set $\mathcal{E}$ of $H$ is the disjoint union of the maximal clique sets of the graph layers: $\mathcal{E} = \bigsqcup_{l=1}^{L} \mathcal{M}(\mathcal{E}_l)$, where $\mathcal{M}(\mathcal{E}_l)$ is the set of all maximal cliques in $\mathcal{E}_l$.

**Graph-superposition decomposition** We build a greedy algorithm to generate a graph-superposition from a hypergraph $H = (\mathcal{V}, \mathcal{E})$ in $\mathcal{O}(3^{|\mathcal{V}|/3} L |\mathcal{E}|)$. The procedure is outlined in Algorithm 1. The generation of the graph-superposition decomposition relies on an algorithm that adds hyperedges sequentially. It starts with an empty graph-superposition representation with a predefined number of layers. This graph-superposition is progressively filled by adding hyperedges one by one in a random order. For each new hyperedge, a random layer is chosen from those that do not interfere with the new maximal clique given by the edge at hand, following Definition 3.4. If the hyperedge cannot be added to any layer, the algorithm fails. After a

predefined number of attempts, the hypergraph is considered non-projectable on the given number of layers.

**Definition 3.4** (Condition to add a hyperedge to a layer)**.** Let $(\mathcal{V}, (\mathcal{E}_l)_{l=1}^{L})$ be a graph-superposition representation and $e$ a hyperedge. For a given layer number $l \in [\![1, L]\!]$, let $\mathcal{M}$ be the set of the maximal cliques in the graph $(\mathcal{V}, \mathcal{E}_l)$ and $\mathcal{M}'$ the set of the maximal cliques in the graph $(\mathcal{V}, \mathcal{E}_l \cup \mathcal{P}_2(e))$. $\mathcal{P}_2(e) \subset \mathcal{V}^2$ is the set of all possible node pairs that are part of hyperedge $e$. The hyperedge $e$ is admissible for addition to the $l$-th layer of the graph-superposition if the following conditions are satisfied:

$$e \notin \mathcal{M} \quad \wedge \quad \mathcal{M}' = \mathcal{M} \cup \{e\} \qquad (1)$$

The condition in Definition 3.4 ensures that adding $e$ creates exactly one new maximal clique, namely $e$, while leaving all previously encoded maximal cliques unchanged. Doing so makes the projection unambiguous: we assign each hyperedge to exactly one layer, and later recover all hyperedges by enumerating the maximal cliques of each layer. Treating a hyperedge as a clique is an encoding choice independent of the structural patterns of the original hypergraph.

**Algorithm complexity** In Figures 9 and 10, we show the time and number of attempts required to perform the graph-superposition decomposition. Results demonstrate that even for $L \ll |\mathcal{V}|$, the majority of graphs are projected in under a second and require at most a handful of attempts. We discuss this in further detail in Appendix D.3. In practice, $L$ can be selected by starting from a small value and increasing it only when repeated projection attempts fail; in our experiments, at most six layers were sufficient.

**Hypergraph reconstruction** Given a graph-superposition, the corresponding hypergraph can be reconstructed with no loss of information using the maximal cliques of each graph layer. The algorithm starts with the empty hypergraph $H = (\mathcal{V}, \emptyset)$. For each layer of the superposition, the set of maximal cliques is $\mathcal{M} = \{\mathcal{E}_1, \dots, \mathcal{E}_m\}$. Each of the cliques $\mathcal{E}_i$ is then used to form a new hyperedge and added to $H$. At the end of this procedure, the reconstructed hypergraph is identical to the one used to form the graph-superposition, up to the ordering of layers and hyperedges. Thus, the full round trip from hypergraph to graph-superposition and back does not introduce any loss of information.

### 3.3. Diffusion via Graph-Superposition Transformer

Since the proposed graph-superposition projection encompasses multiple standard graphs, we require an extended graph diffusion model formulation to generate new samples. Moreover, alongside dependencies between nodes and edges within a single graph, our model must consider dependencies between different representations of the same node across superposition layers, as well as between the graph-level properties of the different layers. Thus, we present the generalized diffusion technique, followed by details of our novel transformer architecture below. Figure 4 shows the diffusion pipeline and the integration of our Graph-Superposition Transformer-based model.

**Forward diffusion** We use discrete graph diffusion (Vignac et al., 2023) as the starting point for our diffusion process. For an unattributed graph with binary edges as input, the classic forward-noising process is modeled as a Markov chain. Within it, the adjacency matrix encoding the edges of a graph is progressively corrupted towards a predetermined prior distribution over a specified number of time steps, based on a noise schedule. Notably, a key feature of discrete diffusion is that the edges in the adjacency matrices at different steps remain discrete, yielding valid graph structures. In our case, given the superposition $(\mathcal{V}, (\mathcal{E}_l)_{l=1}^{L})$, let $(\mathcal{E}_l^t)_{ij} \in \{0, 1\}$ denote the adjacency entry of nodes $i$ and $j$ in superposition layer $1 \leq l \leq L$ at time step $0 \leq t \leq T$. Time step $t = 0$ corresponds to the original graph, while the case $t = T$ corresponds to the fully noised graph. For each layer $l$, we have the following transition probability relationship from the single graph case:

$$
\begin{bmatrix} \Pr\left((\mathcal{E}_l^{t+1})_{ij} = 1\right) \\ \Pr\left((\mathcal{E}_l^{t+1})_{ij} = 0\right) \end{bmatrix}
$$
$$
= \begin{bmatrix} \alpha_t + (1 - \alpha_t)m & (1 - \alpha_t)m \\ (1 - \alpha_t)(1 - m) & \alpha_t + (1 - \alpha_t)(1 - m) \end{bmatrix}
$$
$$
\begin{bmatrix} \Pr\left((\mathcal{E}_l^{t})_{ij} = 1\right) \\ \Pr\left((\mathcal{E}_l^{t})_{ij} = 0\right) \end{bmatrix}.
$$

where $(\mathcal{E}_l^t)_{ij}$ denotes the value of edge $(i, j)$ in the adjacency matrix corresponding to $\mathcal{E}_l^t$, $\alpha_t$ is the retention probability for the transition from time $t$ to time $t+1$ and is set according to the noise schedule, and $m = \Pr_{\text{data}}((\mathcal{E}_l^0)_{ij} = 1)$ denotes the empirical marginal probability of an edge being present, so that $(m, 1-m)$ defines the binary noise distribution. We apply the forward noise process independently to each edge in each superposition layer, keeping the process tractable and efficient during training and sampling.

**Reverse diffusion** The reverse diffusion process iteratively denoises a random sample from the noise distribution without access to the reference clean graph, effectively generating new data points from the modeled distribution. Specifically, we train a neural network $\theta$ to approximate the clean graph from any given time step $t$. We gradually denoise the sampled noise graph during generation, based on $\theta$'s predictions over $T$ steps, to converge toward a realistic graph sample. To train the model, we treat each adjacency matrix entry in each superposition layer as a binary classification problem and thus optimize a cross-entropy objective between the model's predicted probability distribution for each edge and the ground-truth edge values. Specifically,

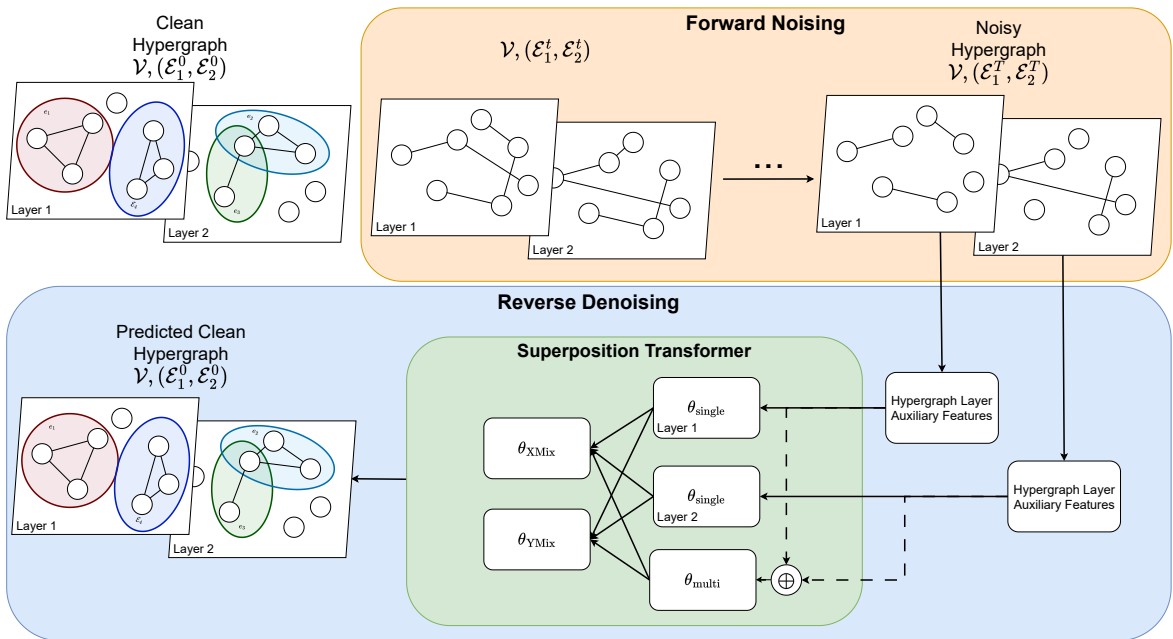

*Figure 4.* SuperHype Diffusion Process and Model Architecture Overview. The *forward noising* procedure takes a graph-superposition representation and iteratively adds noise to it. A *Graph-Superposition Transformer* is trained to *reverse* the diffusion process by taking *layer auxiliary features* as input and returning the predicted clean hypergraph.

for a reference clean graph-superposition $(\mathcal{E}_l^0)_{l=1}^L$, and the prediction $(\hat{\mathcal{E}}_l^0)_{l=1}^L$, we have:

$$\mathcal{L} = \sum_{l=1}^L \frac{2}{\mathcal{V}^2} \sum_{i,j \in |\mathcal{V}|^2} \text{CrossEntropy}((\hat{\mathcal{E}}_l^0)_{ij}, \ (\mathcal{E}_l^0)_{ij})$$

**Graph-Superposition Transformer** To take advantage of the richer representation enabled by the superposition, we must extend the standard graph transformer architecture at the core of many modern graph diffusion models to account for interaction between superposition layers. Since the superposition layers share the same set of nodes, the node representations across layers should inform each other. Similarly, it is beneficial to share graph-level information between the layers. For edges, we do not want to merge edge representations between layers at an input level, as that would destroy one of the main characteristics of the superposition. However, we share the same set of neural network weights for edges across different layers. Doing so enables common patterns across layers to be modeled more efficiently without having to be relearned separately. While we strictly generate structural information for our task, previous discrete diffusion work has shown that adding auxiliary node-, graph-, and edge-level information to the model input significantly improves generation quality Xu et al. (2024); Vignac et al. (2023). Such information is derived directly from the input graph and consists of node degrees, cycle counts, and the total number of connected components. Our modified architecture relies heavily on

such auxiliary structural features, and in Section 3.4 we discuss the hypergraph-specific auxiliary features we use to maximize performance.

Each transformer block in our model incorporates five types of embeddings derived from auxiliary structural features. Edge embeddings also include the layer-specific adjacency value of the corresponding node pair. Figure 4 includes an overview of such a transformer block. We have three layer-specific embeddings: node $\mathbf{X}$, edge $\mathbf{E}$, and graph $\mathbf{Y}$ features. They are applied only to the distinct components within the same layer, and are modeled by a set of parameters $\theta_{\text{single}}$ shared across layers. The two multi-layer embeddings correspond to nodes, $\mathbf{X}^+ = \sum_{l=1}^L \mathbf{X}_l$, and the whole graph, $\mathbf{Y}^+ = \sum_{l=1}^L \mathbf{Y}_l$. We initialize each via the sum of the corresponding embeddings within each layer. We model interaction between $\mathbf{X}^+$ and $\mathbf{Y}^+$ with a set of parameters $\theta_{\text{multi}}$. After processing via the two parallel parameter sets $\theta_{\text{single}}$ and $\theta_{\text{multi}}$, we also mix the updated global node embeddings $\mathbf{X}^+$ with the ones from different layers $(\mathbf{X}_1, \dots, \mathbf{X}_L)$ via another set of parameters $\theta_{\text{XMix}}$. We proceed analogously for graph-level embeddings via a $\theta_{\text{YMix}}$ parameter set.

Our design separates layer-specific and cross-layer reasoning. The shared $\theta_{\text{single}}$ block processes each graph layer with the same parameters, so common local patterns do not have to be relearned independently for each layer, while the edge embeddings remain layer-specific to preserve the clique-layer assignment that makes the superposition unambiguous.

The multi-layer embeddings $\mathbf{X}^+$ and $\mathbf{Y}^+$ are obtained by summing over layers and are therefore invariant to layer permutations. The layer-wise blocks are equivariant to layer permutations, up to the same permutation of their outputs. $\theta_{\text{XMix}}$ and $\theta_{\text{YMix}}$ are the only components not strictly invariant, since their inputs include a concatenated sequence of layer-level representations.

### 3.4. Hypergraph-specific auxiliary features and triplet aggregation

Below, we introduce two optimizations applied to our Graph-Superposition Transformer model.

**Auxiliary structural features** Since the graph-superposition representation focuses on cliques to describe hypergraphs, it is natural to provide the Graph-Superposition Transformer with information about the cliques in the input. We focus on 3-cliques and 4-cliques, as they can be computed efficiently and remain relevant even for hyperedges containing more than four nodes, since multiple 3- and 4-cliques are contained within larger (maximal) cliques.

Within each superposition layer, we attach information about cliques at the node and edge levels (denoting the number of cliques each node or edge belongs to) and at the graph level (denoting the total number of cliques). We include more details in Appendix A.1.

**Triplet aggregation** As covered above, in a clique, every subset of nodes also forms a clique. Thus, smaller cliques, like 3-cliques, are a fundamental building block for larger cliques. In addition to enriching the model input with auxiliary features, we thus tackle the problem directly at the architectural level. Following Hussain et al. (2024), we augment the layer-specific graph transformer module $\theta_{\text{single}}$ with triplet aggregation. The module is applied independently to each superposition layer and enriches intra-layer node-pair message passing with information from node triplets. Triplet aggregation introduces new interaction signals between pairs of nodes based on the third node in their shared 3-cliques. Furthermore, it allows for better modeling of dependencies between non-connected nodes with a common neighbor. Its runtime complexity of $\mathcal{O}(|\mathcal{V}|^{2.37})$ (Vassilevska, 2009) also allows integrating it into the model without creating a bottleneck. For each layer, the triplet-aggregation module updates the representation of every node pair $\{i, j\}$ by aggregating information through all possible intermediate nodes $k$. It first computes an inward message $\mathbf{o}_{ij}^{\text{in}}$ and an outward message $\mathbf{o}_{ij}^{\text{out}}$, which are then combined with an MLP to obtain the triplet-aggregation update for the node pair. The aggregation weights are defined by scalar logits $b_{ik}^{\text{in}}, b_{ki}^{\text{out}}$ and gates $g_{ik}^{\text{in}}, g_{ki}^{\text{out}}$, while the value vectors $\mathbf{v}_{jk}^{\text{in}}$ and $\mathbf{v}_{kj}^{\text{out}}$ are predicted by MLPs. The resulting messages are

computed as follows:

$$\mathbf{o}_{ij}^{\text{in}} = \sum_{k=1}^{N} a_{ik}^{\text{in}} \mathbf{v}_{jk}^{\text{in}}, \quad \mathbf{o}_{ij}^{\text{out}} = \sum_{k=1}^{N} a_{ki}^{\text{out}} \mathbf{v}_{kj}^{\text{out}}$$

$$a_{ik}^{\text{in}} = \texttt{softmax}_k(b_{ik}^{\text{in}}) \times \sigma(g_{ik}^{\text{in}})$$

$$a_{ki}^{\text{out}} = \texttt{softmax}_k(b_{ki}^{\text{out}}) \times \sigma(g_{ki}^{\text{out}})$$

Clique-aware features and triplet interactions also reduce the risk of splitting maximal cliques during denoising: the auxiliary counts expose local clique structure to the model, whereas triplet aggregation encourages consistent edge decisions within node triples.

## 4. Experiments and Results

In this section, we compare our model against several state-of-the-art hypergraph generation methods across different datasets and metrics. We include in our comparison the HYGENE (Gailhard et al., 2025) and HyperPLR (Wen & Yu, 2025) models, as well as the former's baselines.

**Datasets** We evaluate our model on the datasets from Gailhard et al. (2025). Four of them are generated from probabilistic models: Erdős-Rényi, SBM (hypergraphs with a community structure), Hypertree (hypergraphs with a tree structure), and Ego (hypergraphs centered on a single node). The remaining dataset consists of bookshelf objects from ModelNet40 (Wu et al., 2015) converted to hypergraphs.

**Baselines** We compare our model with six baselines. HYGENE is a diffusion-based model from Gailhard et al. (2025) for the generation of small synthetic hypergraphs. We also include the baselines VAE, GAN, and Diffusion, in which the hypergraph incidence matrices are generated directly using different standard deep learning paradigms. HyperPA is another baseline taken from Gailhard et al. (2025), which constructs hypergraphs from a predetermined distribution of node degrees and hyperedge sizes. Finally, HyperPLR is the model from Wen & Yu (2025). It generates hypergraphs using the clique expansion as an intermediate representation.

**Evaluation metrics** In this section, we focus on five main metrics to evaluate our model. Node number, edge size, and node degree metrics are the Wasserstein distances between the distributions of the predicted batch and the reference dataset (Appendix D). See Appendix E for more details on the choice of the reference data split for evaluation. The spectral metric is the quadratic maximum mean discrepancy between the two Laplacian eigenvalue distributions. For SBM, Hypertree, and Ego, we further measure the proportions of valid, unique, and novel hypergraphs among the generated ones. For novelty and uniqueness, we convert each generated and reference hypergraph to its bipartite representation and then apply the standard graph-isomorphism protocol used in graph-generation work (Vignac et al., 2023).

| Model | SBM Hypergraphs $(n_{avg} = 31.73, std = 0.55)$ | | | | | Ego Hypergraphs $(n_{avg} = 109.71, std = 10.23)$ | | | | | Tree Hypergraphs $(n_{avg} = 32, std = 0)$ | | | | |
|---|---|---|---|---|---|---|---|---|---|---|---|---|---|---|---|
| | V.U.N.↑ | Node Num↓ | Node Deg↓ | Edge Size↓ | Spectral↓ | V.U.N.↑ | Node Num↓ | Node Deg↓ | Edge Size↓ | Spectral↓ | V.U.N.↑ | Node Num↓ | Node Deg↓ | Edge Size↓ | Spectral↓ |
| HyperPA | 2.5% | 7.5e-2 | 4.1 | 4.1e-1 | 2.7e-1 | 0% | 3.6e1 | 2.6 | 4.2e-1 | 2.4e-1 | 0% | 2.4 | 3.2e-1 | 2.8e-1 | 1.6e-1 |
| VAE | 0% | 3.8e-1 | 1.3 | 1.1 | 2.4e-2 | 0% | 4.8e1 | 8.0e-1 | 1.5 | 1.3e-1 | 0% | 9.7 | 7.2e-2 | 4.8e-1 | 1.2e-1 |
| GAN | 0% | 1.2 | 2.1 | 1.2 | 5.9e-2 | 0% | 6.0e1 | 9.1e-1 | 1.7 | 2.3e-1 | 0% | 6.0 | 1.5e-1 | 4.7e-1 | 8.9e-2 |
| Diffusion | 0% | 1.5e-1 | 1.7 | 1.4 | 3.1e-2 | 0% | 4.5 | 4.0 | 3.0 | 1.9e-1 | 0% | 2.2 | 1.7 | 1.9 | 1.3e-1 |
| HyperPLR | 0 (0)% | 1.0e1 (1) | 1.4 (3e-2) | **0 (0)** | 4.0e-2 (2.1e-2) | 0% | 1.3e1 | 4.7e-1 | 1.8e-1 | 2.2e-2 | 0 (0)% | 0 (0)* | 0 (0)* | 0 (0)* | 0 (0)* |
| HYGENE | 63% | **7.2e-2** | 1.7 | 3.5e-3 | 2.6e-2 | **95.6%** | 9.3e-1 | 1.5e-1 | 4.5e-1 | 5.6e-3 | 79.2% | **0** | 3.2e-2 | 1.0e-1 | 7.9e-3 |
| Ours | **88.3 (0.7)%** | 1.0e-1 (1e-2) | **6.8e-1 (1.6e-1)** | 2.6e-3 (4e-4) | **3.0e-3 (2e-4)** | 50.6% | **3.1e-1** | **7.9e-2** | **8.0e-3** | **1.3e-3** | **90 (3)%** | 4.0e-3 (1.7e-3) | **3.7e-3 (7e-4)** | **3.2e-3 (4e-4)** | **3.1e-4 (6e-5)** |

*Table 1.* Comparison between SuperHype and baselines for SBM, Ego, and Tree hypergraphs. Values in parentheses represent the standard deviations over three experiments. For HyperPA, VAE, GAN, and Diffusion, we report values from Gailhard et al. (2025). *For HyperPLR on the tree dataset, generated hypergraphs are identical to the training ones.

| Model | Erdos-Renyi Hypergraphs $(n_{avg} = 32, std = 0.07)$ | | | | ModelNet40 Bookshelf $(n_{avg} = 119.38, std = 68.20)$ | | | |
|---|---|---|---|---|---|---|---|---|
| | Node Num↓ | Node Deg↓ | Edge Size↓ | Spectral↓ | Node Num↓ | Node Deg↓ | Edge Size↓ | Spectral↓ |
| HyperPA | **0.000** | 5.5 | 1.8e-1 | 1.8e-1 | 8.0 | 7.6 | 4.4e-2 | 4.8e-2 |
| VAE | 1.0e-1 | 2.1 | 5.4e-1 | 3.5e-2 | 4.7e1 | 6.2 | 1.5 | 1.9e-1 |
| GAN | 6.8e-1 | 2.6 | 6.6e-1 | 4.8e-2 | **0.0** | 4.0e2 | 4.6e1 | 4.8e-1 |
| Diffusion | 5.0e-2 | 2.2 | 7.8e-1 | 1.4e-2 | **0.0** | 2.0e1 | 2.3 | 7.9e-2 |
| HyperPLR | 1.7 (4e-2) | 5.6 (2e-1) | 1.3 (6e-3) | 1.2e-1 (2e-3) | 1.2 | **3.0** | **0** | **1.9e-3** |
| HYGENE | 2.6e-2 | 2.0e-1 | 1.4e-1 | 4.7e-3 | 1.7 | 8.2 | 3.1e-2 | 6.1e-2 |
| Ours | 7.2e-3 (1.3e-3) | **1.5e-1 (3e-2)** | **6.0e-3 (3.6e-3)** | **9.2e-4 (1.3e-4)** | 2.4 | 3.2 | 7.6e-2 | 2.9e-2 |

*Table 2.* Comparison between SuperHype and baselines for the ER and ModelNet40 hypergraphs. Values in parentheses represent the standard deviations over three experiments. For HyperPA, VAE, GAN, and Diffusion, we report values from Gailhard et al. (2025).

**Experimental setup** We run our experiments on NVIDIA H100 NVL and L40S GPUs, with computation times for both training and evaluation ranging from one to five days. We tune the embedding sizes in our model so that these tasks take approximately the same time as those for HYGENE. For HyperPLR, we do not perform such tuning, as it has considerably lower overhead. Our experiments are conducted with 1000 generated hypergraphs, and the metrics are calculated using the training dataset as a reference. For the experiments on HYGENE, we fixed the EMA rate to 0.9999 because there does not seem to be any clear differences between the possible values. In doing so, we avoid selecting the best values of a random process with variable outcomes. For the weaker HyperPA, VAE, GAN, and Diffusion baselines, we instead use the results from (Gailhard et al., 2025) directly, which are calculated over a batch of 40 generated hypergraphs and a reference dataset with the same number of hypergraphs.

**Graph-superposition decomposition** We project the hypergraphs of these five datasets to create the corresponding graph-superposition representations, requiring six layers for SBM, five for Erdős-Rényi, one for Hypertree, three for the Ego, and six for ModelNet40 Bookshelf.

**Results** We summarize our results in Tables 1 and 2. Overall, our model outperforms the baselines in accurately reproducing both the local and global features of the training hypergraphs in the generated batch. In all cases, 100% of generated hypergraphs are not isomorphic to any other generated one, or to any training hypergraph.

The metrics on edge sizes and node degrees provide information on the models' ability to reproduce the local structure of the hypergraphs in the training dataset with precision. Across all datasets with a low number of nodes, our model is better or comparable to HYGENE on this aspect.

On the **SBM**, **Ego** and **Tree** datasets (Table 1), SuperHype consistently outperforms most of the baselines. Notably, SuperHype is always the best method under node degree and spectral distribution. Furthermore, SuperHype generates the most valid graphs on the SBM (88.3%) and Tree (90%) datasets. While on the Ego dataset, SuperHype falls short in terms of validity, it outperforms HYGENE on all other quality metrics. Overall, HyperPLR delivers inconsistent generative quality due to the ambiguity introduced by its clique expansion representation. HYGENE delivers decent generative performance thanks to its extensive bipartite representation. However, it falls behind SuperHype 12 times

out of 15 scenarios.

On the **Erdős-Rényi** dataset (Table 2), SuperHype continues to deliver state-of-the-art generative quality, being the best performing method on four metrics out of five, and second-best in the remaining one. On the larger **Model-Net40** dataset (Table 2), all methodologies struggle to yield consistent quality. Some baselines, like GAN and Diffusion, overfit on some metrics while significantly degrading on others. SuperHype, HyperPLR, and HYGENE are the only methods that deliver consistent performance. Hyper-PLR delivers good generative quality on this dataset, with SuperHype being a close second. Overall, SuperHype consistently delivers state-of-the-art generative performance on all datasets. At the same time, state-of-the-art methods are highly inconsistent.

The spectral and validity metrics are representative of the models' ability to learn and reproduce with precision the global structure of the hypergraphs. In Table 1, it seems that our model outperforms HYGENE in its capability to reproduce valid SBM hypergraphs and hypertrees. In Table 2, the results are still satisfying on the larger hypergraphs of ModelNet40 Bookshelf.

For the Ego dataset, the proportion of valid ego hypergraphs is worse with our model than with HYGENE. Such an outcome arises from representing hyperedges as maximal cliques. If an edge is missing in a set of nodes that is supposed to be a maximal clique representing a hyperedge, the hyperedge is split into two smaller hyperedges, and, in some cases, one of them does not contain the ego node, which creates an invalid Ego hypergraph.

The Ego dataset highlights a failure mode specific to recovering hyperedges as maximal cliques. If the denoising model misses a pairwise edge inside a node set that should form a maximal clique, the corresponding hyperedge can be split into several smaller maximal cliques. In ego hypergraphs, this can create an invalid sample when one of the resulting cliques no longer contains the ego node. Our clique-count auxiliary features and triplet aggregation help mitigate this issue by making the model explicitly aware of local clique structure and of dependencies between node pairs with common neighbors. However, they do not impose a hard combinatorial constraint, and thus cannot fully eliminate split-clique errors.

For the ModelNet40 Bookshelf dataset, none of the tested models appear to produce suitable hypergraphs. Even though the statistics on node number and edge size are fairly good, the node-degree errors for HyperPLR, HYGENE, and SuperHype are too large for the generated hypergraphs to be used as hypergraphs derived from the original dataset. Indeed, the average node degree for the training dataset is 7.13, whereas the error on node degrees is 3.17 for SuperHype

and 8.21 for HYGENE.

**Limitations** Although SuperHype achieves strong generation quality on the tested benchmarks, its current dense implementation is primarily suited to small and medium-sized hypergraphs. In particular, our experiments show good performance for hypergraphs with fewer than roughly 200 nodes, while the computational cost increases sharply with the number of nodes. This scalability boundary is common to current state-of-the-art hypergraph diffusion methods, such as HYGENE, where evaluation spans largely the same datasets. Despite operating on an exact representation, SuperHype has comparable, often lower, training and sampling times than HYGENE in our benchmark setting (see Appendix B).

Leveraging sparsity is a promising route for improved scalability. The graph-superposition decomposes a hypergraph into several simpler graph layers, whose adjacency matrices are usually sparse. Following sparse discrete diffusion approaches (Qin et al., 2025b), future implementations could avoid materializing dense $V^2$ edge representations at each diffusion step by maintaining sparse layer representations, applying sparse multiplications on existing edges, and treating non-edges through aggregate or sampled updates rather than dense all-pairs computations.

Other robustness limitations suitable for future work concern clique splitting and permutation invariance. While auxiliary structural features and triplet aggregation substantially reduce such errors, the current model lacks a recovery mechanism when a generated maximal clique splits. Similarly, although most parts of our architecture are already permutation-invariant, an end-to-end invariant formulation would help mitigate the risk of residual bias from the ordering of superposition layers.

## 5. Conclusion

We introduce SuperHype to address the generation quality issues of existing hypergraph generators that rely on bipartite or weighted clique representations. SuperHype is a novel diffusion model for hypergraphs, which represents hypergraphs exactly and efficiently via a graph-superposition projection. SuperHype maps each hypergraph to a small set of graphs, each with the same number of nodes as the original hypergraph. We design a Graph-Superposition Transformer that learns the global data structure across graph projections over the forward and reverse diffusion processes. Finally, we further enhance the resulting model's performance with auxiliary features and a customized triplet-aggregation mechanism. Our experimental evaluation across five hypergraph datasets shows that SuperHype outperforms state-of-the-art baselines on graph and spectral metrics.

## Impact Statement

Hypergraphs and their synthesis have a broad impact across fields where modeling complex higher-order interactions is relevant, including social networks, epidemics, and chemical reactions. As a generative model, our work is relevant for tasks such as training downstream machine learning models and evaluating novel algorithms on hypergraphs. Notably, using synthetic samples as a stand-in or alongside real ones can help protect the confidentiality of sensitive real data or expand the pool of data available for analysis.

## Acknowledgments

This research is partly funded by the Priv-GSyn (200021E_229204/550302673) and Tracer (2000-1-242997) projects of the Swiss National Science Foundation, as well as the DEPMAT project (P20-22/N21022) of the Dutch Research Council, and ASM International NV. This research is partly funded by NExTWORKx, a collaboration between TU Delft and KPN on future telecommunication networks. This research is partly supported by the DYMAN project funded by the European Union - European Innovation Council under G.A. n. 101161930.

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

# A. Details on the model

Below, we elaborate on the auxiliary features we pass on to our model and provide additional details of the model architecture.

## A.1. Auxiliary features

Before every forward pass in the network, the model computes additional information to deepen the network's understanding of the complex interactions between nodes. The calculated vectors are fed into the different embedding types, which are then transformed into a continuous embedding via a multi-layer perceptron. The following information is provided for each embedding type:

- **e** (edges): The number of 3-cliques and 4-cliques containing the edge

- **Y** (graph-superposition): The number of 3-cliques and 4-cliques in the entire graph-superposition

- **X** (multi-layer node): The number of 3-cliques and 4-cliques containing one of the different versions of the node across the layers

- **x** (single-layer node): The concatenation of the vector X containing multi-layer information and the number of 3-cliques and 4-cliques containing the node in the given layer

- **y** (one layer): The concatenation of the vector Y containing multi-layer information and the number of 3-cliques and 4-cliques in the layer

Note that this auxiliary information covers all 3-cliques and 4-cliques, not merely the maximal ones.

This process relies on calculating, for each layer $l$, the matrices $(B_{i,j})_{1 \leq i,j \leq N}$ and $(C_{i,j})_{1 \leq i,j \leq N}$, which respectively contain the number of 3-cliques and 4-cliques for every edge. The basic algorithm consists of calculating, for every triplet and quadruplet, the product of the potential edge values in the adjacency matrix:

$$\forall (i,j,l) \in [\![1,N]\!]^2 \times [\![1,L]\!], \begin{cases} B_{i,j} = \sum_{k=1}^{N} A_{i,k} A_{k,j} A_{j,i} \\ C_{i,j} = \sum_{k=1}^{N} \sum_{p=1}^{N} A_{i,k} A_{k,j} A_{j,i} A_{i,p} A_{k,p} A_{j,p} \end{cases} \tag{2}$$

Another approach to these calculations is to account for the sparsity of the adjacency matrix to avoid computing all edge products. The following algorithm reduces the number of operations by skipping pairs of nodes that have been detected as not being connected:

In Algorithm 2, the loops are stopped when two nodes in the potential clique are detected as not being connected, which means that the node set has no chance of being a clique.

In the early stages of the denoising process, the distribution of edges lacks a precise structure, which can sometimes yield an enormous number of 3- and 4-cliques that do not provide the network with usable information about the hypergraph's structure. Therefore, we define a maximum above which the value is replaced with $-1$. In our settings, this value is 100 for edges, 1000 for nodes, and 10,000 for the entire network.

This formulation does not use hard combinatorial constraints, even though it could, in principle, further improve generation quality, especially for large hyperedges. From experimentally observed hyperedge-size and node-degree metrics, we find that our hypergraph-specific auxiliary features are nevertheless effective at guiding the model to form cliques. Furthermore, we can theoretically demonstrate that a small number of erroneous edges in the graph-superposition decomposition yields a limited difference in the final number of hyperedges. Take a graph with a single $N$-node maximal clique. If we remove a single edge from this clique, we get two maximal cliques of size $N - 1$. Now, let $G$ be an arbitrary graph with an edge $e$ contained in $C$ maximal cliques. Also, let $G'$ be the graph $G$ without edge $e$. Based on the previous reasoning, we can upper bound the difference between the number of maximal cliques $M(G)$ in $G$ and the corresponding $M(G')$ in $G'$ as:

$$|M(G) - M(G')| \leq C$$

---

**Algorithm 2** Optimized version of the algorithm to compute the number of 3-cliques and 4-cliques for every edge in a graph.

---

**Input:** Adjacency matrix $(A_{i,j})_{1 \leq i,j \leq N}$ of the $l$-th layer of the graph-superposition

**Output:** Matrices $(B_{i,j})_{1 \leq i,j \leq N}$ and $(C_{i,j})_{1 \leq i,j \leq N}$ denoting the edge-wise number of 3-cliques and 4-cliques, respectively.

$(B_{i,j})_{i,j} \leftarrow (0)_{i,j}$
$(C_{i,j})_{i,j} \leftarrow (0)_{i,j}$
**for** $i \leftarrow 1$ **to** $N$ **do**
    **for** $j \leftarrow i+1$ **to** $N$ **do**
        **if** $A_{i,j} = 0$ **then**
             Break
        **end**
        **else**
            **for** $k \leftarrow j+1$ **to** $N$ **do**
                **if** $A_{i,k} = 0 \vee A_{j,k} = 0$ **then**
                     Break
                **end**
                **else**
                    $B_{i,j} \leftarrow B_{i,j} + 1$
                    **for** $p \leftarrow k+1$ **to** $N$ **do**
                        **if** $A_{i,p} = 1 \wedge A_{j,p} = 1 \wedge A_{k,p} = 1$ **then**
                            $C_{i,j} \leftarrow C_{i,j} + 1$
                        **end**
                    **end**
                **end**
            **end**
        **end**
    **end**
**end**
**return** $(B_{i,j})_{1 \leq i,j \leq N}$ and $(C_{i,j})_{1 \leq i,j \leq N}$

---

A similar reasoning can apply to an edge addition. Thus, even if the denoising process makes a few isolated errors on the edges, the result still corresponds to a hypergraph with a reasonable number of hyperedges.

### A.2. Details on the architecture

This section details the network used to denoise the graph-superposition representation. Figure 5 shows the connections between the different modules of the model.

Figure 6 describes the cross-attention module between local and global embeddings. It corresponds to the XYTransformer ($\theta_{\text{multi}}$ in the main text), the XxTransformer ($\theta_{\text{XMix}}$ in the main text), and the YyTransformer ($\theta_{\text{YMix}}$ in the main text). In all of these cases, the variable L corresponds to the embedding of the local feature, which is a refinement of the global feature G. In the XYTransformer, the local feature is X and the global feature is Y. In the XxTransformer, the local feature is x and the global feature is X. In the YyTransformer, the local feature is y, and the global feature is Y.

Figure 7 shows the layout of the graph transformer module, which realizes message-passing between layer-specific embeddings. First, it computes pairwise interactions to update node, edge, and layer embeddings. After this, triplet aggregation coefficients are used to update the edge embeddings.

## B. Complexity analysis

In the following, we estimate the asymptotic complexity of our proposed model, making use of the notation below:

- $N$: number of nodes

- $L$: number of layers in the graph-superposition representation

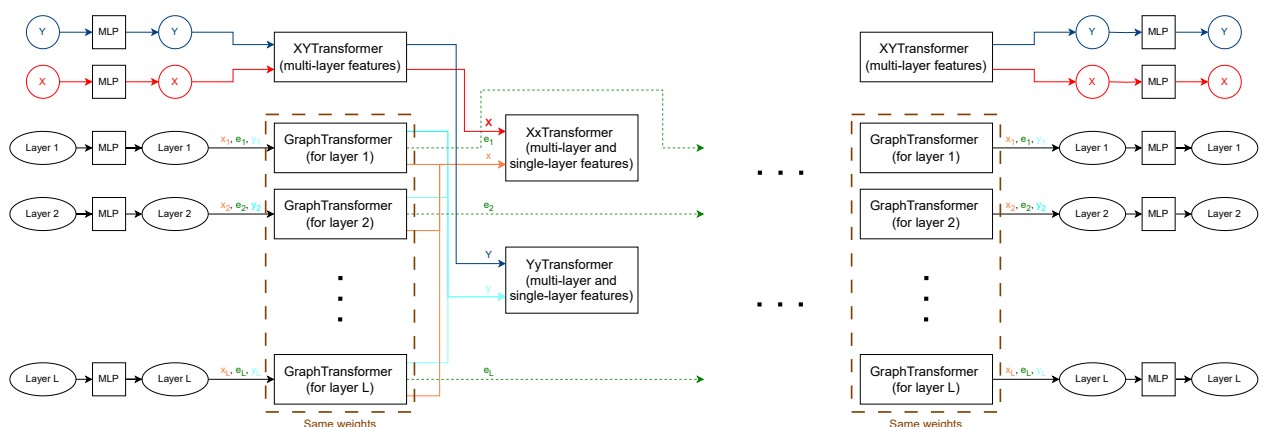

*Figure 5.* The complete architecture of the Graph-Superposition Transformer.

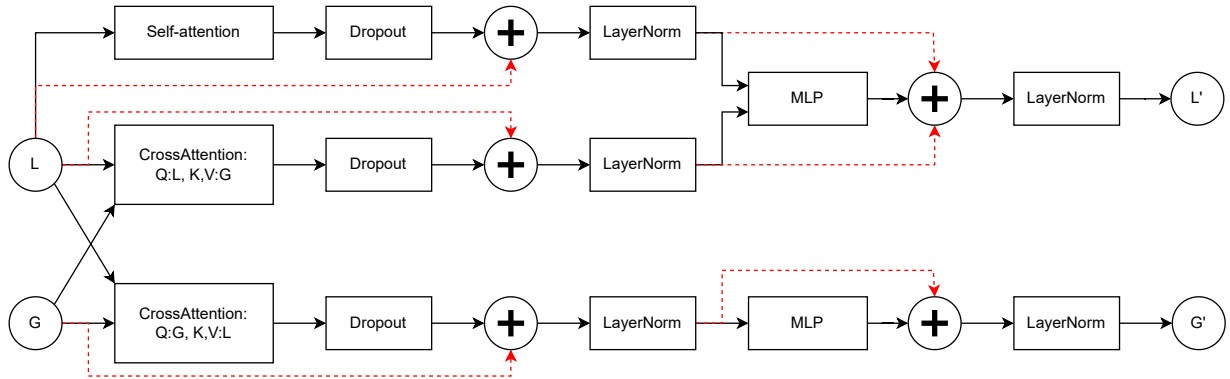

*Figure 6.* The cross-attention module between local features (L) and global features (G).

- $d$: dimension of the embeddings ($d = \max(d_x, d_e, d_y, d_X, d_Y)$)

We calculate the complexity relative to a batch containing a single hypergraph represented with a superposition of $L$ graphs with $N$ nodes and embeddings of size $d$. This complexity applies to both a training step and a generation run. We consider every module of the Graph-Superposition Transformer separately.

First, the graph-transformer block enables message passing between layer-specific node embeddings ($x$), edge embeddings ($e$), and graph embeddings ($y$). It contains linear modules on $x$, $e$, and $y$, with an asymptotic complexity of $O(N^2 L d)$ because there are $L$ layers with $N^2$ edge embeddings of size $d$ on each. These linear modules compute queries, keys, and values, as well as the modulation factor coefficients in feature-wise linear modulation (FiLM). The query-key product $\frac{QK}{\sqrt{d_f}}$ also has a cost of $O(N^2 L d)$, as well as the weighted sum of the attention head, the statistical pooling, or the FiLM on $e$ and $x$.

For the triplet aggregation, we only detail the inward case; the outward update has the same complexity. It relies on a two-step process:

- Calculation of the coefficients $a_{ik}^{\text{in}} = \text{softmax}_k(b_{ik}^{\text{in}})\sigma(g_{ik}^{\text{in}})$

- Aggregation of these coefficients $o_{ij}^{\text{in}} = \sum_{k=1}^{N} a_{ik}^{\text{in}} v_{jk}^{\text{in}}$

The first step has an asymptotic complexity of $O(N^2 L d)$. The second step is expressible as a matrix product:

$$O = AV^T$$

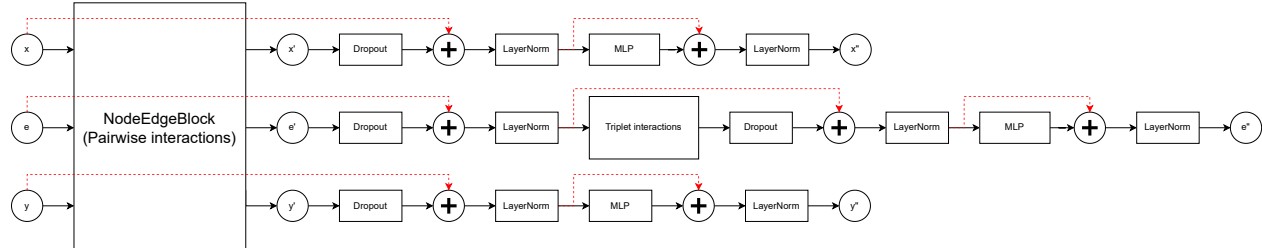

*Figure 7.* The integration of triplet aggregation inside the graph transformer module.

According to Hussain et al. (2024), this matrix product has a complexity of $\approx O(dN^{2.37})$ for a single layer, which gives a complexity of $\approx O(LdN^{2.37})$ for the entire graph-superposition.

For the attention layer between $X$ and $Y$, the attention is calculated between $N$ multi-layer node embeddings and a global embedding, all of them of size $d$, which gives an asymptotic complexity of $O(N^2 d)$. For the attention layer between $X$ and $x$, the attention is calculated, for each of the $N$ multi-layer node embeddings, between the multi-layer node embedding and the $L$ corresponding single-layer node embeddings, which gives a complexity of $O(L^2 N d)$. For the attention layer between $Y$ and $y$, the attention is calculated between a global embedding and $L$ single-layer graph embeddings, which gives a complexity of $O(L^2 N)$.

Before and after the transformer modules, there is, for every type of embedding, a multi-layer perceptron to convert between the input/output representation and embeddings. If the ratio between the hidden dimensions and the dimensions of the embeddings is bounded, the complexity of these layers is $O(N^2 L d)$.

Before passing the graph superposition to the network, the auxiliary information has to be calculated. Appendix A.1 provides two possible algorithms. We discuss the complexity of a single layer of the graph-superposition representation with $N$ nodes, $m_2$ edges, and $m_3$ 3-cliques per algorithm.

The first basic algorithm consists of computing the sums in Equation (2) using the product of matrix elements. This operation is easy to parallelize and, for this reason, it is GPU-friendly. However, the complexity is $O(N^3)$ for the 3-cliques and $O(N^4)$ for the 4-cliques.

The second algorithm stops the nested loops when a pair of nodes in the set does not correspond to an edge. For the indices $i$ and $j$, all possibilities are explored, but the index $k$ is examined only if there is an edge between $i$ and $j$, which happens $m_2$ times. Then, if a 3-clique is detected, every possible fourth node is examined to check if it forms a clique with the three previous ones. For every 3-clique, these operations are in $O(N)$. Therefore, the complexity of the second algorithm for cliques is $O(N(N + m_2 + m_3))$ per layer and $O(NL(N + m_2 + m_3))$ for the entire graph-superposition.

Consequently, the global complexity of the Graph-Superposition Transformer is $\approx O(LdN^{2.37})$ without the auxiliary features, $\approx O(LdN^{2.37} + N^4 L)$ with the first algorithm for the auxiliary information, and $\approx O(LdN^{2.37} + NL(N + m_2 + m_3))$ with the second one. Although the asymptotic complexity of the auxiliary information calculation is high, this process is often much faster than a forward pass in a neural network.

## C. Intermediate experimental results

We conducted experiments to compare our model with different architectural variants. The results for the variants are calculated from a set of 200 generated hypergraphs, which are compared to the training dataset; the results for our model are from the main experiments with 1000 generated hypergraphs.

Table 3 shows a comparison of the accuracy with different types of auxiliary features on the SBM dataset. Three variants are tested: a model without auxiliary features, a model with the auxiliary features used in Vignac et al. (2023) (cycle count and spectral auxiliary features), and our auxiliary features with cliques of sizes 3 and 4. The configuration without auxiliary data seems to be significantly less accurate than the others in all aspects of the generated hypergraphs. Regarding the type of auxiliary features, there is no clear difference between those of Vignac et al. (2023) and ours regarding the proportion of valid SBM, the node degrees, and the Laplacian eigenvalues. However, clique-specific auxiliary features achieve higher accuracy for hyperedge size.

| Experiment | Valid SBM | Node degree | Edge size | Spectral |
|---|---|---|---|---|
| No auxiliary feature | 0% | 0.558 | 0.0028 | 0.0072 |
| Cycle + spectral auxiliary features | 84% | 0.270 | 0.00352 | 0.00303 |
| Clique auxiliary features | 87.4% | 0.276 | 0.00273 | 0.00239 |

*Table 3.* Comparison of the accuracy of the Graph-Superposition Transformer on the SBM dataset with different kinds of auxiliary features.

| Experiment | Valid SBM | Node degree | Edge size | Spectral |
|---|---|---|---|---|
| Graph transformer on multi-hot encoded edge-labeled graphs | 68.5 % | 0.78 | 0.034 | 0.0091 |
| Graph-Superposition Transformer without triplet interaction | 77% | 0.563 | 0.0108 | 0.00419 |
| Graph-Superposition Transformer with triplet aggregation | 87.4% | 0.276 | 0.00273 | 0.00239 |
| Graph-Superposition Transformer with triplet attention | 86% | 0.416 | 0.00297 | 0.00294 |

*Table 4.* Comparison of different kinds of transformer architectures with clique auxiliary features on the SBM dataset.

Table 4 shows a comparison of the accuracy of different variants of our architecture on the SBM dataset. The classic graph transformer, when used with edge labels, yields lower accuracy across all metrics, confirming the need for a transformer model specifically designed for the topology of a graph-superposition. Adding any form of triplet aggregation improves the model's accuracy across all metrics for this dataset. However, triplet attention, which requires heavier computations, does not seem to improve the model's accuracy compared to triplet aggregation, a lighter variant. For this reason, we decided to use triplet aggregation.

## D. Experimental details

### D.1. Datasets

We tested our model on both synthetic and non-synthetic hypergraphs. These are the datasets used by Gailhard et al. (2025). Except for the Tree hypergraph dataset, which contains 1000 hypergraphs for training, 100 for validation, and 100 for testing, these datasets contain 128 hypergraphs for training, 32 for validation, and 40 for testing.

**Erdős-Rényi hypergraphs**: This is the most basic way to generate random hypergraphs. It starts with a predefined set $S \subset \mathbb{N}^*$ of possible hyperedge sizes, and a family of probabilities $(p_i)_{i \in S}$. For a combination of $i \in S$ nodes in the hypergraphs $H = (\mathcal{V}, \mathcal{E})$, their probability to form a hyperedge is given by $p_i$:

$$\forall i \in S, \forall e \in \mathcal{P}_i(S), \mathbb{P}(e \in \mathcal{E}) = p_i$$

The dataset that is used is generated with $S = \{2; 3; 4\}$ and $p_2 = 0.1, p_3 = 0.005, p_4 = 0.0005$.

**Tree hypergraphs**: First, a random tree with 32 nodes is sampled following a uniform distribution. Then, connected edges are randomly merged to form hyperedges with up to five nodes.

**SBM hypergraphs**: First, each of the 32 nodes is assigned to one of the two categories with independent and identically distributed Bernoulli processes of probability 0.5. Then, for every combination of three distinct nodes, the corresponding hyperedge is added to the hypergraph with probability 0.05 if all nodes are in the same community, and with probability 0.001 if they are in different communities.

**Ego hypergraphs**: First, a random number $n$ of nodes is chosen following a uniform distribution on $[\![150, 200]\!]$. Then 3000 random hyperedges are added with the following process: for each hyperedge, a random number $k$ of nodes is chosen uniformly in $[\![2, 5]\!]$. Then a random combination of $k$ nodes is chosen uniformly in $\mathcal{P}_k(\mathcal{V})$. Once all of the hyperedges have been added, a node is chosen uniformly in $\mathcal{V}$ to be the ego one. Finally, all of the hyperedges that do not contain the ego node are removed.

In all synthetic hypergraphs, even though there is a theoretical number of nodes, any isolated node is removed from the

generated hypergraph. Figure 8 provides an example for each type of synthetic hypergraph used in our experiments.

**ModelNet40 hypergraphs**: These datasets are constructed by converting the ModelNet40 datasets (Wu et al., 2015) into hypergraph datasets for the classes *bookshelf*, *piano*, and *plant*. These datasets contain only hyperedges of size three.

### D.2. Description of the metrics

The precision of our model and the baselines was evaluated using the metrics from Gailhard et al. (2025). Since HyperPA relies on a different generation process with a given distribution of node degrees and edge sizes for each hypergraph, we use the results of Gailhard et al. (2025) with only 40 generated hypergraphs. We also retain the results from Gailhard et al. (2025) for the Diffusion, VAE, and GAN baselines, as their poor precision made it difficult to compute the metrics.

**Node degree**: The Wasserstein distance between the node degree distribution of the reference dataset and that of the generated batch. These distributions are constructed by separately adding each node's degree to a distribution common to all hypergraphs. It is not a comparison of the node degree distributions across separate hypergraphs, but rather a comparison of the node degree distributions across the entire datasets.

**Edge size**: The Wasserstein distance between the edge size distribution of the reference dataset and that of the generated batch. As for node degrees, this is the distribution of the edge sizes across the entire dataset.

**Spectral**: The square of the Maximal Mean Discrepancy of the Laplacian eigenvalue distribution between the reference dataset and the generated batch.

**Node number**: For our experiments, it is the Wasserstein distance between the node number distributions of the reference dataset and the generated batch. For the values taken from Gailhard et al. (2025), it is the average difference between the target number of nodes and the number of nodes in the generated hypergraph.

**Validity metric for Tree hypergraphs**: The Tree hypergraphs are created by merging adjacent edges in a tree graph. They can be characterized by the fact that the only cycles in the clique expansion are hyperedge subsets. The aforementioned property is tested to compute the proportion of valid Tree hypergraphs in the generated batch.

**Validity metric for SBM hypergraphs**: SBM hypergraphs consist of two node communities, with intra- and inter-community probabilities for forming a hyperedge. To determine whether a hypergraph is a valid SBM, it is first projected to a graph via clique expansion. Then, an optimal community distribution is found. The hypergraph is considered a valid SBM if it has two communities and the intra- and extra-community probabilities match those used to generate the training dataset.

**Validity metric for ego hypergraphs**: Since an ego hypergraph is a hypergraph containing one node belonging to all of the hyperedges, testing this property enables us to compute the proportion of valid ego hypergraphs in the generated batch.

**Closeness centrality, betweenness centrality and harmonic centrality**: These metrics come from Aksoy et al. (2020). They are calculated with $s = 1$.

**Uniqueness**: Proportion of generated hypergraphs that are not isomorphic to another generated one.

**Novelty**: Proportion of generated hypergraphs that are not isomorphic to any hypergraph in the training dataset.

### D.3. Multi-layer graph projection of hypergraphs in practice

As mentioned in Section 3.2, our algorithm is not guaranteed to create a graph-superposition from a hypergraph. Indeed, if a conflict is detected between the maximal cliques of a layer, meaning they no longer represent the hyperedges of the input hypergraph, the algorithm fails and is retried with a different random seed until success. Using this process, we obtain a relatively low projection time compared to the model training time.

In Algorithm 2, each *MaximalCliques* call uses a variant of the Bron and Kerbosch algorithm (Tomita et al., 2006), and has a time complexity of $O(3^{N/3})$. For $L$ layers and $N$ nodes, the total time complexity for the superposition projection is $O(3^{N/3}LN)$. However, most graphs in practice have significantly fewer maximum cliques than the theoretical maximum and are sparse, drastically reducing runtime (Eppstein et al., 2013). The max clique algorithm is shown to scale to tens of thousands of nodes (Tomita et al., 2006), leaving the denoising model, not projection, as the bottleneck of our system. The histograms in Figure 9 and Figure 10 show the runtime and the number of attempts required for a successful projection across different datasets and layer counts.

Although our stochastic algorithm does not guarantee projection within the requested number of layers on the first attempt, we observe that most graphs are projected in under a second and require at most a handful of attempts. Only a small fraction of graphs require many attempts. Finally, when running Erdős-Rényi with a greater number of target layers, the number of attempts needed decreases as expected.

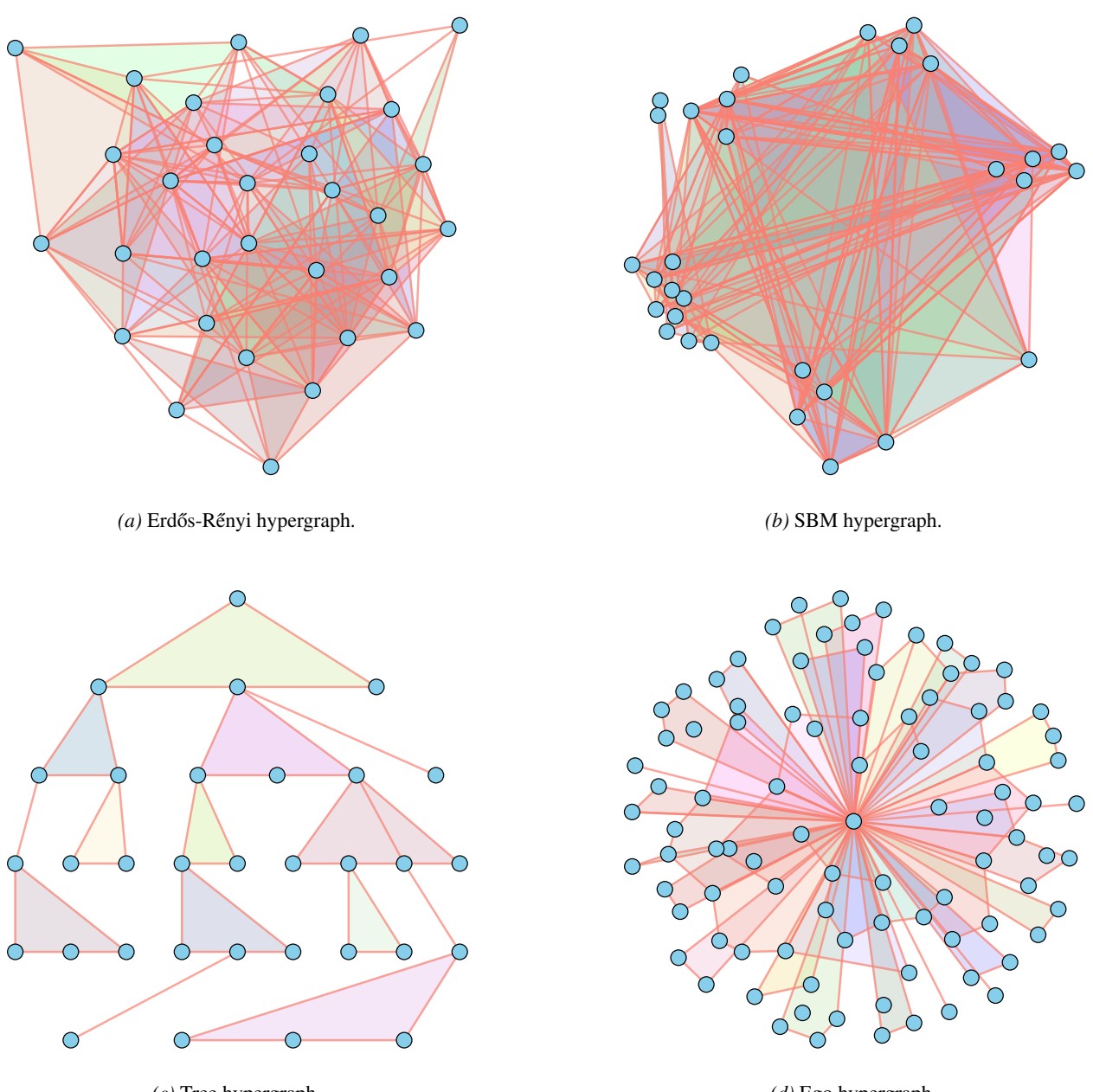

*(a)* Erdős-Rĕnyi hypergraph.

*(b)* SBM hypergraph.

*(c)* Tree hypergraph.

*(d)* Ego hypergraph.

*Figure 8.* Examples of hypergraphs from the synthetic datasets.

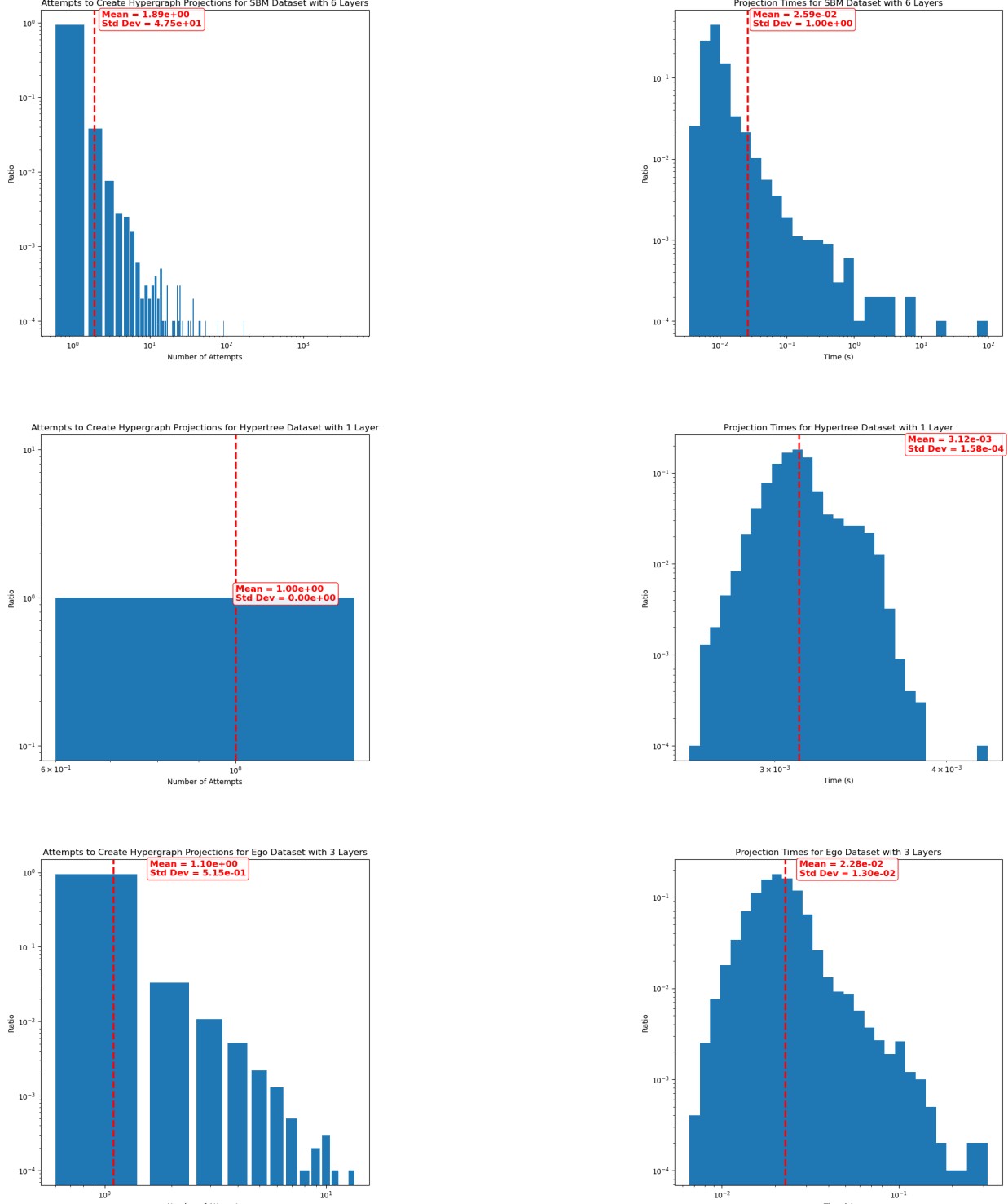

*Figure 9.* Histograms of projection attempts and runtimes for each of the 10 000 hypergraphs sampled from the distributions of SBM (six layers), Ego (three layers), and Tree hypergraphs (one layer) datasets.

| Model | Erdős-Rényi Hypergraphs ($n_{avg}$ = 32, $std$ = 0.07) | | | | | | | | | SBM Hypergraphs ($n_{avg}$ = 31.73, $std$ = 0.55) | | | | | | | | | |
|---|---|---|---|---|---|---|---|---|---|---|---|---|---|---|---|---|---|---|---|
| | Node Num ↓ | Node Deg ↓ | Edge Size ↓ | Spectral ↓ | Uniq. ↑ | Nov. ↑ | Cent. Close ↓ | Cent. Betw. ↓ | Cent. Harm. ↓ | Valid SBM ↑ | Node Num ↓ | Node Deg ↓ | Edge Size ↓ | Spectral ↓ | Uniq. ↑ | Nov. ↑ | Cent. Close ↓ | Cent. Betw. ↓ | Cent. Harm. ↓ |
| HyperPA | **0.0** | 5.5 | 1.8e-1 | 1.8e-1 | 1 | 1 | 7.8e-2 | 1.4e-2 | 1.1e2 | 2.5% | 7.5e-2 | 4.1 | 4.1e-1 | 2.7e-1 | 1 | 1 | 7.4e-2 | 8.0e-3 | 7.8e1 |
| VAE | 1.0e-1 | 2.1 | 5.4e-1 | 3.5e-2 | 1 | 1 | 7.9e-2 | 8.0e-3 | 1.4e1 | 0% | 3.8e-1 | 1.3 | 1.1 | 2.4e-2 | 1 | 1 | 7.0e-3 | 6.0e-3 | 6.5 |
| GAN | 6.8e-1 | 2.6 | 6.6e-1 | 4.8e-2 | 1 | 1 | 1.0e-1 | 1.1e-2 | 1.7e1 | 0% | 1.2 | 2.1 | 1.2 | 5.9e-2 | 1 | 1 | 7.6e-2 | 1.2e-2 | 1.1e1 |
| Diffusion | 5.0e-2 | 2.2 | 7.8e-1 | 1.4e-2 | 1 | 1 | 4.8e-2 | 3.0e-3 | 1.2e1 | 0% | 1.5e-1 | 1.7 | 1.4 | 3.1e-2 | 1 | 1 | 4.0e-2 | 4.0e-3 | 1.4e1 |
| HyperPLR | 1.7 (4e-2) | 5.6 (2e-1) | 1.3 (1e-2) | 1.2e-1 (2e-3) | 1 (0) | 1 (0) | 1.4e-3 (3e-3) | **2.0e-7 (8e-9)** | 28 (9e-1) | 0 (0)% | 1.0e1 (1) | 1.4 (3e-2) | **0 (0)** | 4.0e-2 (2.1e-2) | 9.8e-1 (1e-2) | 1 (0) | 7.2e-2 (1.0e-2) | **2.1e-7 (2e-8)** | 1e1 (1) |
| HYGENE | 2.6e-2 | 2.0e-1 | 1.4e-1 | 4.7e-3 | 1 | 1 | 2.2e-2 | 5.6e-4 | 4.2 | 63% | **7.2e-2** | 1.7 | 3.5e-3 | 2.6e-2 | 1 | 1 | 1.6e-2 | 7.8e-3 | 1.2e1 |
| Ours | 7.2e-3 (1.3e-3) | **1.5e-1 (3e-2)** | **6.0e-3 (3.6e-3)** | **9.2e-4 (1.3e-4)** | 1 (0) | 1 (0) | **1.3e-3 (8e-4)** | 2.1e-4 (6e-5) | **1.2 (2e-1)** | **88.3% (0.7%)** | 1.0e-1 (1e-2) | **6.8e-1 (1.6e-1)** | 2.6e-3 (4e-4) | **3.1e-3 (2e-4)** | 1 (0) | 1 (0) | **5.1e-3 (3.2e-3)** | 2.2e-3 (7e-4) | **4.9 (1.3)** |

*Table 5.* Detailed evaluation metrics for Erdős-Rényi and SBM hypergraphs. The values in parentheses represent the standard deviations over three experiments.

| Model | Ego Hypergraphs ($n_{avg}$ = 109.71, $std$ = 10.23) | | | | | | | | | | Tree hypergraphs ($n_{avg}$ = 32, $std$ = 0) | | | | | | | | | |
|---|---|---|---|---|---|---|---|---|---|---|---|---|---|---|---|---|---|---|---|---|
| | Valid Ego ↑ | Node Num ↓ | Node Deg ↓ | Edge Size ↓ | Spectral ↓ | Uniq. ↑ | Nov. ↑ | Cent. Close ↓ | Cent. Betw. ↓ | Cent. Harm. ↓ | Valid Tree ↑ | Node Num ↓ | Node Deg ↓ | Edge Size ↓ | Spectral ↓ | Uniq. ↑ | Nov. ↑ | Cent. Close ↓ | Cent. Betw. ↓ | Cent. Harm. ↓ |
| HyperPA | 0% | 3.6e1 | 2.6 | 4.2e-1 | 2.4e-1 | 1 | 1 | 3.5e-1 | 2.0e-3 | 1.4e2 | 0% | 2.4 | 3.2e-1 | 2.8e-1 | 1.6e-1 | 1 | 1 | 4.8e-1 | 1.7e-1 | 5.9 |
| VAE | 0% | 4.8e1 | 8.0e-1 | 1.5 | 1.3e-1 | 1 | 1 | 5.6e-1 | 1.9e-2 | 3.9e1 | 0% | 9.7 | 7.2e-2 | 4.8e-1 | 1.2e-1 | 1 | 1 | 2.8e-1 | 1.4e-1 | 3.9 |
| GAN | 0% | 6.0e1 | 9.2e-1 | 1.7 | 2.3e-1 | 1 | 1 | 6.1e-1 | 1.5e-2 | 4.2e1 | 0% | 6.0 | 1.5e-1 | 4.7e-1 | 8.9e-2 | 1 | 1 | 2.0e-1 | 1.2e-1 | 2.2 |
| Diffusion | 0% | 4.5 | 4.0 | 3.0 | 1.9e-1 | 1 | 1 | 4.1e-1 | 9.0e-3 | 6.9 | 0% | 2.2 | 1.7 | 1.9 | 1.3e-1 | 1 | 1 | 3.5e-1 | 1.4e-1 | 8.6 |
| HyperPLR | 0% | 1.3e1 | 4.7e-1 | 1.8e-1 | 2.2e-2 | 1 | 1 | 1.3e2 | **2.0e-7** | 5.4e2 | **100% (0%)** | **0 (0)** | **0 (0)** | **0 (0)** | **0 (0)** | 1 (0) | 0 (0) | **0 (0)** | **0 (0)** | **0 (0)** |
| HYGENE | **95.6%** | 9.3e-1 | 1.5e-1 | 4.5e-1 | 5.6e-3 | 1 | 1 | **1.2e-2** | 1.7e-2 | 3.4 | 79.2% | **0** | 3.2e-2 | 1.0e-1 | 7.9e-3 | 1 | 1 | 2.0e-2 | 1.7e-2 | 5.1e-1 |
| Ours | 50.6% | **3.1e-1** | **7.9e-2** | **8.0e-3** | **1.3e-3** | 1 | 1 | 1.5e-2 | **3.4e-4** | **1.6** | 90 (3.3)% | 4.0e-3 (1.7e-3) | 3.7e-3 (7e-4) | 3.2e-3 (4e-4) | 3.1e-4 (6e-5) | 1 (0) | 1 (0) | 6.4e-3 (3.2e-3) | 5.7e-3 (3.1e-3) | 9.3e-2 (3.3e-2) |

*Table 6.* Detailed evaluation metrics for Ego and Tree hypergraphs. The values in parentheses represent the standard deviations over three experiments.

## D.4. Additional results

In Tables 5 to 7, we present the complete results of the experiments.

In Table 8, we present the training/sampling time and the maximum admitted batch size for HyperPLR, HYGENE, and SuperHype over different datasets. We use the maximum batch size supported by each combination of model and dataset on our hardware as a measure of memory efficiency. We see that SuperHype is generally faster and consumes less memory than Gailhard et al. (2025), the closest, but still inferior, baseline in terms of quality. Wen & Yu (2025) is the most efficient model overall, but its generation quality is much lower than that of both other models.

# E. Motivation for calculating metrics with respect to training data

In Gailhard et al. (2025), the authors compute distances between the generated distribution and the distribution of a testing dataset. Even though both the training and test datasets are sampled from the same probability distribution, the distributions of values such as node degree and edge size are not exactly the same.

If the distances between the sampled batch and the reference batch are much greater than those between the training and testing datasets, calculating the metrics with respect to either dataset does not significantly change the final result. If not, the choice of the reference dataset will have a strong impact on the final metrics. Unlike Gailhard et al. (2025), we chose the training dataset.

First, if the training and testing distributions are too different, even though they are sampled from the same distribution, it

| Model | ModelNet40 Bookshelf ($n_{avg}$ = 119.38, $std$ = 68.20) | | | | | | | | |
|---|---|---|---|---|---|---|---|---|---|
| | Node Num ↓ | Node Deg ↓ | Edge Size ↓ | Spectral ↓ | Uniq. ↑ | Nov. ↑ | Cent. Close ↓ | Cent. Betw. ↓ | Cent. Harm. ↓ |
| HyperPA | 8.0 | 7.6 | 4.4e-2 | 4.8e-2 | 1 | 1 | 2.1e-1 | 5.0e-3 | 8.8e2 |
| VAE | 4.7e1 | 6.2 | 1.5 | 1.9e-1 | 1 | 1 | 1.5e-1 | 3.0e-3 | 1.1e2 |
| GAN | **0.0** | 4.0e2 | 4.6e1 | 4.8e-2 | 1 | 1 | 7.1e-1 | 7.0e-3 | 6.7e2 |
| Diffusion | **0.0** | 2.0e1 | 2.3 | 7.9e-2 | 1 | 1 | 2.4e-1 | 6.0e-3 | 2.6e2 |
| HyperPLR | 1.2 | **3.0** | **0** | **1.9e-3** | 1 | 1 | **2.8e-3** | **3.2e-7** | **5.6e1** |
| HYGENE | 1.7 | 8.2 | 3.1e-2 | 6.1e-2 | 1 | 1 | 8.0e-2 | 4.3e-3 | 2.0e2 |
| Ours | 2.4 | 3.2 | 7.6e-2 | 2.9e-2 | 1 | 1 | 1.6e-1 | 5.9e-3 | 9.0e1 |

*Table 7.* Detailed evaluation metrics for ModelNet40 Bookshelf.

| Dataset | Model | Sampling time (h:m:s) | Training time (h:m:s) | Max batch size |
|---------|-------|----------------------|----------------------|----------------|
| SBM | HYGENE | 00:42:20 | 19:00:00 | 20 |
|  | HyperPLR | 00:00:08 | 00:02:12 | 128 |
|  | SuperHype | 03:20:39 | 09:33:20 | 32 |
| Erdős-Rěnyi | HYGENE | 01:11:37 | 23:00:00 | 20 |
|  | HyperPLR | 00:00:14 | 00:02:29 | 128 |
|  | SuperHype | 02:45:28 | 01:30:00 | 32 |
| Tree | HYGENE | 00:40:00 | 16:45:00 | 20 |
|  | HyperPLR | 00:00:04 | 00:01:27 | 128 |
|  | SuperHype | 00:27:24 | 27:53:26 | 200 |
| Ego | HYGENE | 05:26:40 | 16:00:00 | 10 |
|  | HyperPLR | 00:00:25 | 00:08:11 | 128 |
|  | SuperHype | 06:02:10 | 19:18:20 | 8 |

*Table 8.* Sampling/training time and maximum batch size with the SBM, Erdős-Rěnyi, Tree, and Ego datasets and HyperPLR, HYGENE, and SuperHype.

means the random process by which the testing dataset is sampled from that distribution has a significant impact on the final metrics. Indeed, the model is trained to match a probability distribution learned from the training dataset, so the larger the distance between the training and testing datasets, the greater the distance between the testing dataset and the predicted batch is expected to be. Selecting the training dataset as a reference helps circumvent this random bias in validation metrics.

For example, in synthetic hypergraphs, hyperedges have a probability of existence that depends on their size and position. The model is expected to learn this distribution from the training dataset. It has no way to know the true distribution, so it tries to reproduce the distribution from the training dataset. In this situation, comparing the predicted distribution with the distribution from the training dataset is the only correct way to measure the model's ability to perform this task.

Most of the time, having a test dataset to evaluate a model helps determine whether it is overfitting. It makes sense if we want to get the binary cross-entropy on a denoising task, but for generation, the model does not use any more data from any dataset. In that case, choosing a separate testing dataset does not provide any additional information about the model itself. However, it allows comparing the bias introduced by the dataset choice with that introduced by the model's imprecision. To ensure there is no overfitting, we added metrics for uniqueness and novelty to the generated batch.

On Table 9, distances between the training and the testing dataset are compared to the ones between the testing dataset and the sampled batch. Using these distances, the triangle inequality yields upper and lower bounds on the distances between the training dataset and the sampled batch. In most cases, the distance between the training and testing datasets accounts for most of the errors in validation metrics when computed with respect to the testing dataset. It shows the need for a metric that truly reflects the model's precision, unaffected by random perturbations.

| Metric | Measurement | Erdős-Rényi | SBM | Tree | Ego |
|---|---|---|---|---|---|
| | Distance between train and test | 0.0579 | 0.167 | 0.0215 | 0.0667 |
| | Predicted value for HYGENE | 0.475 | 0.321 | 0.059 | 0.063 |
| Node deg. Wasserstein dist. | Percentage of error | 12.2% | 52.0% | 36.4% | 105.9% |
| | Minimal value for HYGENE | 0.4171 | 0.154 | 0.0375 | 0 |
| | Maximal value for HYGENE | 0.5329 | 0.488 | 0.0805 | 0.1297 |
| | Distance between train and test | 0.00655 | 0.0 | 0.0211 | 0.0522 |
| | Predicted value for HYGENE | 0.012 | 0.002 | 0.108 | 0.220 |
| Edge size Wasserstein dist. | Percentage of error | 54.6% | 0% | 19.6% | 23.7% |
| | Minimal value for HYGENE | 0.00545 | 0.002 | 0.0869 | 0.168 |
| | Maximal value for HYGENE | 0.01855 | 0.002 | 0.1291 | 0.272 |
| | Distance between train and test | 0.00433 | 0.00486 | 0.00547 | 0.00119 |
| | Predicted value for HYGENE | 0.006 | 0.010 | 0.012 | 0.004 |
| Laplacian eigenvals. MMD | Percentage of error | 72.2% | 48.6% | 45.6% | 29.8% |
| | Minimal value for HYGENE | 0.000135 | 0.000917 | 0.00127 | 0.000827 |
| | Maximal value for HYGENE | 0.0205 | 0.0288 | 0.0337 | 0.00955 |

*Table 9.* Comparison of the distances between a batch generated with Gailhard et al. (2025) and a reference dataset and distances between the training dataset and the reference one.

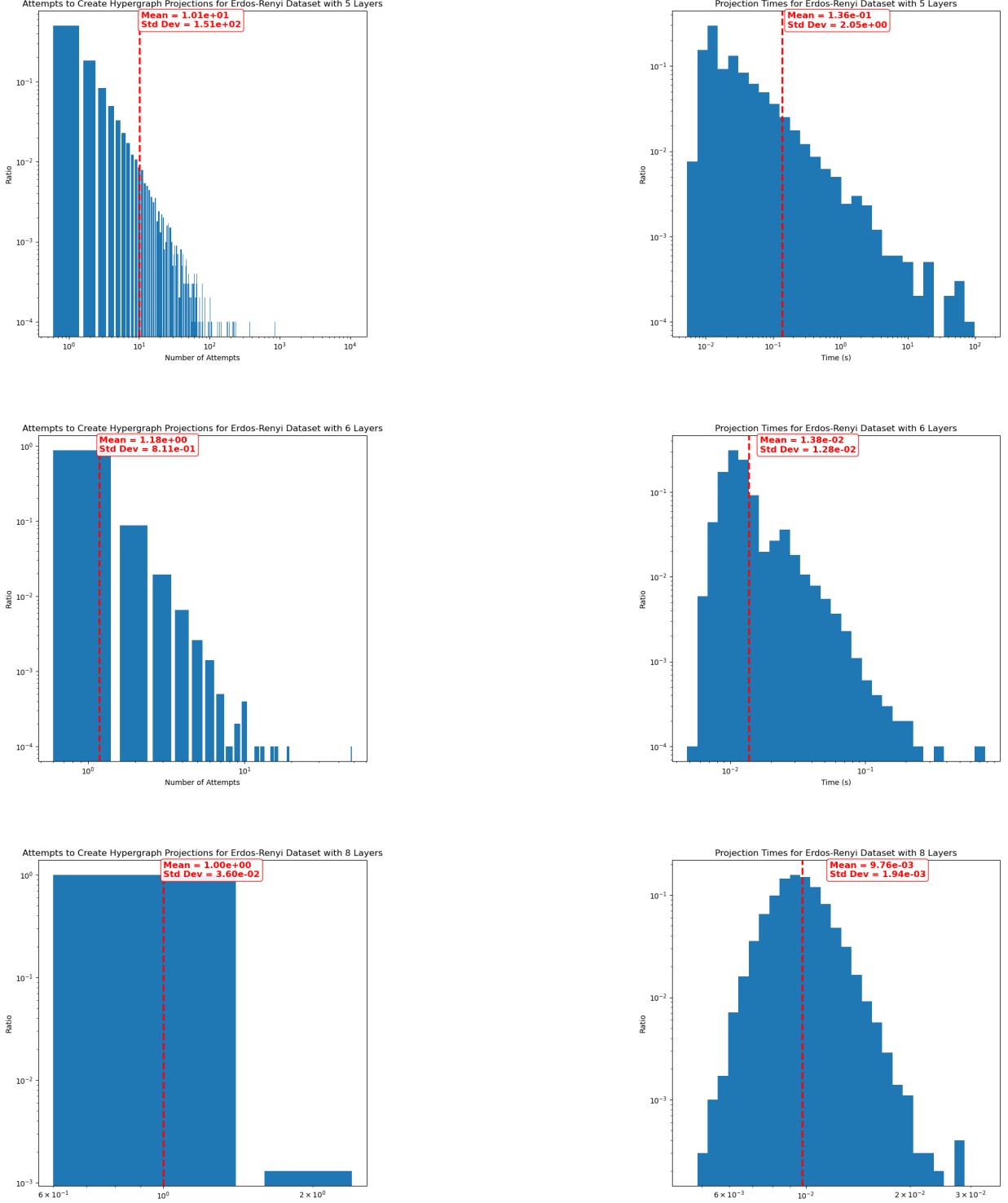

*Figure 10.* Histograms of projection attempts and runtimes for each of the 10 000 hypergraphs sampled from the Erdős-Rényi dataset with five, six, and eight layers.

