# OpenReview forum: "SuperHype: Hypergraph Generation via Graph-Superposition Decomposition"
_ICML.cc/2026/Conference — ICML 2026 regular_

### Official Review · Reviewer_sWpr · 2026-03-06

**Soundness:** 3
**Presentation:** 2
**Significance:** 3
**Originality:** 3
**Overall Recommendation:** 4
**Confidence:** 4

**Summary:**

This paper proposes SuperHype, a hypergraph generation framework based on a new graph-superposition decomposition. The main idea is to represent a hypergraph as a multilayer graph structure and then use a transformer-based discrete diffusion model to generate new samples. The method can also incorporate hypergraph-specific auxiliary features and triplet aggregation to better capture higher-order structure. I find this work promising, but I have several concerns.

**Compliance With Llm Reviewing Policy:**

Affirmed.

**Final Justification:**

The authors have addressed the concerns raised in my initial review through their rebuttal. I maintain a positive evaluation and recommend a Weak Accept (4).

**Key Questions For Authors:**

A. Intuition and justification of the method.

  1. How do the authors justify the consistency of the method? Are there any theoretical insights? Are the generated samples close in distribution to the real ones, and if so, in what sense? This does not need to be a complete theoretical analysis, but some intuition or discussion based on toy examples would be helpful, in addition to the empirical study.

  2. The authors mention the benefit of using a transformer, but can they be more specific? How does the transformer capture interactions between superposition layers, and are there alternative designs?

B. Literature and additional baselines.

  1. The method operates directly on a discrete state space. In contrast, one line of work on hypergraph generation uses a latent-space strategy to avoid working directly in the discrete space, which can be computationally efficient. For example, the authors note that a hypergraph can be represented in bipartite form. Along this line, Wu et al. (2025) study hypergraph generation by connecting the hypergraph to a continuous latent embedding space and training latent diffusion models, while Ma et al. (2026) further extend this approach to hypergraphs with attributes. A related strategy also appears in Vahdat et al. (2021) with a different variational training framework. This line of work is not discussed in the paper.

  2. Latent-space methods are efficient for handling such discrete structure, but they typically rely on low-dimensional assumptions for efficient hypergraph generation. By contrast, methods that operate directly on the discrete state space, while potentially slower in training and sampling (as well as in graph-superposition decomposition in this work), do not rely on such assumptions. In other words, these different strategies may be suitable for different settings. This literature should be discussed and, if possible, included in the empirical comparison rather than omitted.

**References**

- Wu, S., Yang, J., Xu, G., and Zhu, J. (2025). *Denoising Diffused Embeddings: A Generative Approach for Hypergraphs*. arXiv:2501.01541.
- Ma, F., Wu, S., Xu, G., and Zhu, J. (2026). *ReLaSH: Reconstructing Joint Latent Spaces for Efficient Generation of Synthetic Hypergraphs with Hyperlink Attributes*. ICLR 2026.
- Vahdat, A., Kreis, K., and Kautz, J. (2021). *Score-based Generative Modeling in Latent Space*. NeurIPS 34: 11287–11302.

**Limitations:**

See Questions.

**Strengths And Weaknesses:**

**Strengths**
- This paper introduces a novel and well-motivated representation for hypergraphs. The overall framework is coherent, combining the new representation with a transformer-based diffusion model for the hypergraph generation task.
- The ability to incorporate additional modeling components such as auxiliary features and triplet aggregation is appealing.
- The empirical evaluation includes experiments on multiple datasets against several baselines.

**Weaknesses**
- The paper needs to provide more intuition for and justification of the proposed method.
- The literature review and baseline comparisons could be expanded.

See the Questions section for more details on these weaknesses.

---

> ### Author Rebuttal · Authors · 2026-03-31
>
> We are thankful to the reviewer for the insightful feedback and relevant resources. Below is our reply to the reviewer's comments. We will integrate the extra discussion and suggested references in the next revision of the paper.
>
> **W1/Q1: Justify method consistency**
>
> The motivation for our superposition-based approach is two-fold. Firstly, existing state-of-the-art hypergraph generation models operate on inexact representations, meaning that transforming hypergraphs to the representation used by the generation model and back to the original space is inherently lossy (as also exemplified in the answer to Reviewer fqY5's Q1). Secondly, normally a hypergraph is exactly represented as a graph via a bipartite mapping (where one set of nodes corresponds to the hypergraph's nodes, and the other to its hyperedges), which has a large theoretical complexity (as discussed in the answer to Reviewer fqY5's W3), making its use in a diffusion model challenging. Consequently, we propose the superposition representation and build a superposition transformer around it as a more efficient, exact representation for generation. The main insight is that we can decompose an $N$-node hypergraph into a small number of $N$- node graphs. Since the superposition representation is exact, transforming a graph to its superposition and back does not incur any information loss (see also the answer to Reviewer fqY5's W3). Meanwhile, for the actual generation process, we use our diffusion model to jointly generate the layers corresponding to a hypergraph, and then construct the final hypergraph. For evaluation, we use multiple metrics common to previous work in the same setting to measure the statistical similarity of different properties (like edge size and node degree) between real hypergraphs and generated ones across different datasets.
>
> **W1/Q2: Justify transformer benefits**
>
> We follow the trend in the graph generation literature [1, 2] of using graph transformer-based models for denoising, due to their ability to better capture long-range interactions between node pairs than classical message-passing-based layers. Moreover, since our graph superposition effectively entails handling multiple related graphs (that share the same set of nodes), we extend a classical graph transformer block into a superposition transformer. Specifically, instead of computing (node X, edge E, and graph Y) embeddings for a single graph, we compute them for each layer and additionally maintain multi-layer global embeddings for the nodes ($\mathbf{X}^+$) and the whole graph ($\mathbf{Y}^+$) by summing corresponding features across all layers. To preserve the fundamental characteristic of the superposition (where specific cliques in specific layers uniquely represent hyperedges), edge representations are strictly not merged across layers. However, the model shares parameter weights ($\theta_\text{single}$) across layers, allowing it to learn common structural patterns efficiently. Crucially, we add specialized cross-attention modules (XYTransformer, XxTransformer, and YyTransformer) to mix layer-specific and multi-layer features. Doing so ensures that the representations of the same node across layers inform each other. Finally, we extend the attention mechanism with the triplet aggregation of Hussain et al. [3], allowing us to directly model interactions among node triplets rather than only pairs.
>
> **W2/Q1+Q2: Latent diffusion comparison**
>
> We appreciate the suggestion and the useful resources. We will expand our related work section in the next revision of the paper with references to additional hypergraph generation settings, including attributed generation scenarios such as ReLaSH [4], where latent diffusion is critical for jointly modeling hyperlink attributes alongside the hypergraph structure. Our tackled setting involves purely modelling the structural topology of hypergraphs, and our superposition representation is composed of a joint set of related unattributed graphs. Thus, we are operating strictly within a discrete data space and elect to follow the findings from previous works in the classical graph diffusion space (e.g., DiGress [1] and DeFoG [5]), showing that performing discrete diffusion in the data-space allows maximizing the output quality of the generated graph, and importantly helps with maintaining complex properties like graph-level validity (which we measure in Table 1).
>
> _References:_
>
> [1] Vignac et al., DiGress: Discrete Denoising diffusion for graph generation. ICLR 2023.
>
> [2] Jo et al., Graph Generation with Diffusion Mixture. ICML 2024.
>
> [3] Hussain et al., Triplet Interaction Improves Graph Transformers. ICML 2024.
>
> [4] Ma et al., ReLaSH: Reconstructing Joint Latent Spaces for Efficient Generation of Synthetic Hypergraphs with Hyperlink Attributes. ICLR.
>
> [5] Qin et al., DeFoG: Discrete Flow Matching for Graph Generation. ICML 2025.

---

> > ### Author Rebuttal · Reviewer_sWpr · 2026-04-01
> >
> > Thank you for the detailed responses and for giving serious consideration to these questions. Please include these discussions in your revised manuscript to better highlight the contribution and rationale of the methodology and to position the work within the broader literature, including latent space approaches.

---

> > > ### Author Response · Authors · 2026-04-06
> > >
> > > We thank the reviewer for the positive and swift feedback.
> > > We will integrate the above discussion in the next revision of the paper.

---

### Official Review · Reviewer_m267 · 2026-03-10

**Soundness:** 4
**Presentation:** 3
**Significance:** 4
**Originality:** 4
**Overall Recommendation:** 5
**Confidence:** 3

**Summary:**

This paper introduces SuperHype, a novel discrete diffusion model designed for the generation of synthetic hypergraphs. The authors address the inherent NP-complete complexity of hypergraph representation by proposing a graph-superposition projection, an algorithm that embeds a hypergraph into a tractable multi-layer graph representation without any loss of information or ambiguity. To effectively process this decomposed structure during the diffusion and denoising phases, the study introduces a Graph-Superposition Transformer that facilitates message passing both within individual graph layers and globally across the different layers in the superposition. Furthermore, the model is enhanced with a triplet aggregation mechanism and hypergraph-specific auxiliary features, such as 3-clique and 4-clique counts, to better guide the formation of maximal cliques and model higher-order structural dependencies. The main contributions include the exact and tractable graph-superposition representation, the specialized transformer architecture, and extensive empirical evaluations demonstrating that SuperHype consistently outperforms current state-of-the-art baselines across multiple datasets in reproducing both local connectivity patterns and global topological features.

**Compliance With Llm Reviewing Policy:**

Affirmed.

**Key Questions For Authors:**

**1. Automated Heuristic for Hyperparameter $L$:** Could the authors elaborate on whether there is an automated or standardized heuristic to estimate the optimal number of layers ($L$) for completely unseen hypergraphs in real-world applications, rather than relying heavily on trial and error?

*(Impact on evaluation: Clarifying this mechanism would significantly strengthen my confidence in the system's practical usability and deployment readiness, which could positively influence my rating for the paper's Soundness and Significance.)*

**2. Scalability and Sparse Diffusion:** Given the explicitly acknowledged computational bottleneck for hypergraphs exceeding 200-250 nodes, have the authors conducted any preliminary experiments or theoretical explorations regarding the integration of sparse matrix techniques during the early steps of the forward diffusion process?

*(Impact on evaluation: Providing even early empirical evidence or a brief theoretical discussion showing that sparsity could mitigate these costs would greatly enhance my view of the framework's long-term impact and scalability, further solidifying my positive assessment.)*

**3. Robustness against Clique Splitting:** For topologies like the Ego dataset, where generated noise occasionally leads to missing edges and split maximal cliques (resulting in a slight drop in validity), could the authors consider discussing potential lightweight, ambiguity-free post-processing heuristics or targeted loss modifications to seamlessly reconnect these cliques?

*(Impact on evaluation: Proposing a feasible mitigation strategy—or even a conceptual discussion of one—would address the primary empirical weakness regarding generation robustness. This would positively influence my overall assessment of the model's reliability and technical soundness.)*

**Limitations:**

Yes

**Strengths And Weaknesses:**

**Strengths:**

- **Originality & Significance (Novel Representation Design):** The framework offers an elegant, mathematically sound approach by introducing the "graph-superposition decomposition." By circumventing the exponential memory costs of bipartite representations and the inherent ambiguity of standard clique expansions, the method provides a highly practical utility and a significant architectural advantage for complex hypergraph modeling.
- **Originality & Soundness (Architectural Clarity):** The authors' design of the Graph-Superposition Transformer offers a fresh perspective on processing multi-layer graphs in diffusion models. The integration of triplet aggregation to explicitly guide the formation of maximal cliques demonstrates strong technical soundness and aligns perfectly with the proposed projection theory.
- **Significance (High-Fidelity Generation):** The system introduces a reliable pipeline for generating synthetic hypergraphs. The experiments show that the model effectively preserves both local and global topological features. This level of precise reconstructability has a broad impact, making it highly attractive for domains requiring complex relationship modeling.
- **Presentation:** The submission is clearly written, well-structured, and the overall narrative is easy to follow. The methodology, especially the mathematical admissibility conditions for the projection algorithm, is properly contextualized and cleanly explained.

**Weaknesses (Areas for improving Soundness & Presentation):**

- **Soundness (Opportunities for Scalability Enhancements):** The paper provides a thorough empirical evaluation and candidly acknowledges the computational constraints for hypergraphs exceeding 200-250 nodes. The soundness and future impact of the work could be further strengthened by briefly discussing potential theoretical directions—such as integrating sparse matrix techniques during the early diffusion steps—to help scale this elegant framework to larger networks.
- **Soundness (Guidance on Hyperparameter L):** The experimental design currently relies on empirically determined values for the number of layers ($L$). While perfectly suitable for the evaluated datasets, exploring or discussing a more standardized heuristic for estimating $L$ on completely unseen data would provide readers with a more holistic view of the system's deployment process and strengthen the methodological rigor.
- **Soundness (Robustness of Clique Formation):** As the authors elegantly analyze, noise can occasionally lead to missing edges, which might split the maximal cliques (e.g., slightly affecting validity in certain topologies like Ego). Exploring lightweight, ambiguity-free post-processing heuristics to seamlessly reconnect these split cliques could be a fruitful direction to further improve the technical robustness of the pipeline.
- **Presentation (Broader Baseline Context):** The paper provides a solid comparison against strong existing baselines (e.g., HYGENE, HyperPLR). However, considering the rapid evolution of generative models, a slightly expanded qualitative discussion on how this method might adapt to or compare with other emerging efficient graph generation paradigms would make the core claims even more compelling.

**Overall Conclusion:**

Overall, the paper addresses an important problem (Significance) with a highly practical, unambiguous, and creative mathematical approach (Originality). The suggestions mentioned above mainly present opportunities for future scalability and operational heuristics (Soundness) rather than detracting from the core, elegant design of the proposed superposition system. I lean towards acceptance.

---

> ### Author Rebuttal · Authors · 2026-03-31
>
> We wish to thank the reviewer for the positive feedback. We will add all extra discussion/clarifications from our reply to the revised manuscript.
>
> **W1/Q2: Scalability (sparse diffusion)**
> We agree that scaling beyond 250 nodes remains an important step. However, SOTA hypergraph diffusion methods like HYGENE also limit evaluations to these sizes due to high-order relational complexity. Appendix Table 8 shows our training and sampling times are already often lower than HYGENE's. Regarding your excellent sparse matrix suggestion, we expand our theoretical discussion below, building on Qin et al. [1] (noted in Section 4). Since our projection decomposes dense hypergraphs into separate, simpler graph layers, their individual adjacency matrices are naturally sparse. Like DiGress [2], our forward diffusion preserves sparsity using a noisy distribution based on prior layer edge density. Thus, integrating sparse multiplications lets us update only non-zero transition probabilities instead of computing dense $O(N^2)$ matrices per layer. Furthermore, our Transformer's attention modules could leverage sparse mechanisms (e.g., passing messages only along existing edges and sampled non-edges). These enhancements would significantly reduce memory footprints, crucially aiding scaling to thousands of nodes.
>
> **W2/Q1: Hyperparameter L tuning**
> The main factors in determining the required number of layers L are the hyperedge density in the training data and the ratio of nested hyperedges (i.e., hyperedges that are strict subsets of others in the same graph). As such, they can be used to directly derive a reasonable value for L, possibly with minimal further tuning, albeit a possibly higher one than achievable with specific tuning. Still, given that the time to project a graph is short (often much less than a second, as shown in Figures 9 and 10) and the number of required layers is fairly low (at most 6 for our tested datasets), tuning L with a simple automated heuristic can often also remain a valid option. Namely, starting with L=2 we try to project the training hypergraphs to graph superpositions. For any hypergraph that we cannot project within a set number of attempts (e.g., 5), increment L and restart processing the dataset. A single digit L was sufficient in all our experiments. Note that, once L is large enough to allow reliable projection, further increasing its value would only increase overhead, but not output quality.
>
> **W3/Q3: Clique splitting robustness**
> While we take steps to minimize clique splitting, it indeed remains a risk. Specifically, our auxiliary structural features and triplet aggregation make the model aware of generated cliques, highly limiting the number of splitting errors. However, in the current formulation, there is no recovery mechanism to correct these errors when they occur. While we have architected an error-correction mechanism, we chose not to include it in the final model, as it added extra overhead and ultimately amounted to a trade-off in generation quality. Namely, we first train the model to predict, in addition to the graph-superposition representation, information about the number of hyperedges encoded for every node/edge. We then apply a heuristic to add small corrections to the graph-superposition representation in order to match the node and edge labels. However, this procedure increases the complexity of the training process, as it additionally requires optimizing around the model's predicted labels, and it does not improve the generation quality under equal training budgets.
>
> **W4: Presentation**
> We will expand the discussion of related models to include alternative generation paradigms for classic graphs (e.g., hierarchical approaches [3, 4], efficient diffusion[1, 5]). We will additionally mention state-of-the-art solutions for domain-specific hypergraph generation (e.g., hypergraphs with hyperlink attributes [6]).
>
> _References:_
>
> [1] Qin et al.: SparseDiff: Sparse Discrete Diffusion for Scalable Graph Generation. TMLR 2025.
>
> [2] Vignac et al.: DiGress: Discrete Denoising diffusion for graph generation. ICLR 2023.
>
> [3] Jang et al.: Graph Generation with K2-trees. ICLR 2024.
>
> [4] Karami: HiGen: Hierarchical Graph Generative Networks. ICLR 2024.
>
> [5] Chen et al.: Efficient and Degree-Guided Graph Generation via Discrete Diffusion Modeling. ICML 2023.
>
> [6] Ma et al.: ReLaSH: Reconstructing Joint Latent Spaces for... Hypergraphs with Hyperlink Attributes. ICLR 2026.

---

> > ### Author Rebuttal · Reviewer_m267 · 2026-04-06
> >
> > I thank the authors for their detailed rebuttal. The responses have adequately addressed my initial questions and provided helpful clarifications.
> >
> > 1. Scalability and Sparse Diffusion: The theoretical discussion regarding the integration of sparse matrix techniques is clear. Building on Qin et al. and DiGress to explain how the forward diffusion can preserve sparsity, and noting how the Transformer's attention modules can update non-zero transition probabilities rather than dense $O(N^2)$ matrices, addresses my concerns about computational bottlenecks for larger networks.
> >
> > 2. Automated Heuristic for Hyperparameter $L$: The clarification on the factors determining $L$ (hyperedge density and nested hyperedges) is helpful. The proposed automated heuristic---starting at $L=2$ and iteratively incrementing it upon projection failure---provides a reasonable and practical solution that reduces the reliance on manual trial-and-error.
> >
> > 3. Robustness against Clique Splitting: I appreciate the transparency regarding the clique splitting issue. The explanation that an error-correction heuristic was explored but ultimately excluded due to the trade-off with training complexity and overhead is a reasonable justification. Acknowledging this limitation and the bounds of the current formulation in the final text would be beneficial.
> >
> > 4. Presentation (Broader Context): Including alternative hierarchical approaches and efficient diffusion paradigms in the related work will help provide a more comprehensive context for the proposed method.
> >
> > Overall, the rebuttal is satisfactory and addresses the main points raised in my review. I will maintain my current score (Accept). I recommend incorporating these clarifications into the final version of the manuscript.

---

> > > ### Author Response · Authors · 2026-04-06
> > >
> > > We thank the reviewer once more for the positive feedback on the manuscript and rebuttal.
> > > As mentioned in the rebuttal, we will incorporate the rebuttal clarifications in the following manuscript version.

---

### Official Review · Reviewer_fqY5 · 2026-03-12

**Soundness:** 3
**Presentation:** 4
**Significance:** 3
**Originality:** 3
**Overall Recommendation:** 4
**Confidence:** 3

**Summary:**

This paper proposes the SuperHype framework which is a diffusion model for hypergraphs. The authors use the projection of graph superposition to embed the hypergraph into a multi-layer graph embedding (a novel construction in the paper). They also propose a Graph-Superposition Transformer which is used to generate new samples from these hypergraph representations. In total, their framework is able to beat SOTA baselines in terms of reproducing local and global connectivity patterns. The main contributions are threefold: (i) superposition decomposition; (ii) Graph-Superposition Transformer; and (iii) hypergraph-specific auxiliary features and triplet aggregation. This is a substantial contribution because it enables tractable and unambiguous hypergraph representation, for a previously intractable problem.

**Compliance With Llm Reviewing Policy:**

Affirmed.

**Final Justification:**

Thanks to the authors for further explanations. I will maintain my positive evaluation of the paper.

**Key Questions For Authors:**

1. If clique expansion results in information loss, is there any implication for using the maximal clique sets for creation of the graph superposition representation?

2. Proof that the greedy algorithm is optimal for mapping the hypergraph to graph superposition representation? Is there a theoretical bound on how far this greedy approach is from the minima' number of layers required to represent the hypergraph? Did the authors try other algorithms?

3. It is stated that `Given a graph-superposition, the corresponding hypergraph can be reconstructed with no loss of information using the maximal cliques of each graph layer,’ but is there any information loss in constructing the graph-superposition representation of the original graph?

4. It is stated that `In all cases we got 100% of generated hypergraphs that are not isomorphic to any other generated one, or to any hypergraph from the training dataset,’ details as to how is this determined are missing (non-trivial as graph isomorphism problem is in NP; WL-algorithm?)?

5. Does the use of maximal cliques introduce an inductive bias since it would assume that the hyperedges are fully connected components in the projected layers? What happens if the real-world hypergraph does not naturally decompose into cliques?

6. SuperHype framework divided the hypergraph into layers. However, transformers and diffusion models are sensitive to how data is ordered. Is the set of layers treated as an unordered set or an ordered sequence? If the order of the layers in the superposition is shuffled, does the Graph-Superposition Transformer produce the same result? If the model assumes an order, it may fall short in capturing the full symmetry of the hypergraph, limiting its representation. How is layer-permutation invariance handled?

**Limitations:**

Yes.

**Strengths And Weaknesses:**

Strengths:

1. Novel framework for making hypergraph synthesis (which is an NP-complete problem) tractable using graph superposition projection to embed a hypergraph into a multi-layer graph without loss of information.

2. Extensive evaluation across six models (including VAEs, Autoregressors, Gan-based models, and other diffusion models) and five datasets.

3. This paper is also novel in introducing the graph-superposition transformer which is used to enable graph diffusion which then produces high fidelity and high-quality hypergraphs. They also include a triplet-aggregation mechanism which allows for more control over the synthesis process, controlling maximal clique-formation and improving the hypergraph quality.

4. Solid review of existing approaches and where they fall short. Helps to motivate the problem and demonstrate the niche SuperHype is filling.

Weaknesses:

1. No standard deviations for any experiments other than their method. Also, the average and standard deviations are only calculated over three runs. Results would be more convincing with 10 seeds, and average and standard deviation metrics across all baseline methods.

2. Poor scalability to large graphs due to computation cost increase, does not out-perform baselines in this case. Authors note that for graphs with more than 250 nodes, the model will have to be adapted. The authors argue that SuperHype has substantial impact for use in modeling interactions including those in social networks or epidemics – both of which would have scalability issues, rendering SuperHype non-applicable for practical use.

3. Lack of baselines against models using bipartite graph representations which are also lossless. Why are models which use this approach not used as baselines for comparison against another lossless approach? This may be a more fair comparison than baselining only against clique expansion methods which have information loss.

Minor fixes:

- The introduction would benefit from a more thorough explanation of hypergraphs/ motivation for why they are useful.

- Should have more references in the introduction for areas where hypergraphs are used. Many domains are listed but there are only two citations. Also missing names of and citations for SOTA models in the introduction.

---

> ### Author Rebuttal · Authors · 2026-03-31
>
> We would like to thank the reviewer for their comprehensive feedback.
> Our responses to the raised points are below.
> We will ensure that all answers to the reviewer's questions and discussion of weaknesses are included in the next revision of our manuscript.
>
> **W1: No standard deviation for baselines**
>
> We include standard deviations for SuperHype specifically to show better the statistical significance of our performance improvements relative to the baselines and to demonstrate the stability of our model across seeds.
> To maximize the precision of individual measurements, we calculate the metrics for each run over 1000 synthetic graphs, compared to HYGENE's 40.
> Given the high number of graphs per run, we deemed a factor of three sufficient to demonstrate that our model behaves predictably across runs.
> We also note that both our strongest baselines, HYGENE and HyperPLR, do not include standard deviation in their evaluations.
> They rely on the fact that most of their evaluation metrics (like ours) are node- or hyperedge-level, meaning that the number of observed samples to calculate the metrics over is considerably larger than the number of synthetic/real test graphs.
>
> **W2: Scalability**
>
> Most graph synthesizers still face challenges in scaling to hypergraphs with many hundreds of nodes.
> State-of-the-art baselines like HYGENE face similar scalability challenges to our model and limit their evaluation to the same types of datasets.
> Despite performing diffusion over an exact representation (higher overhead, but better generation quality), which is not the case for HYGENE, SuperHype achieves comparable (often lower) training and sampling times, as shown by Table 8 in the appendix.
> Many real-world cases combine complex patterns with large scales, so, despite scalability remaining a challenge in such scenarios, we note in the manuscript that SuperHype's ability to better model complex patterns within benchmark networks (e.g., SBM validity) is an essential part of tackling such workloads.
> As highlighted in our reply to Reviewer m267's Q2, integrating techniques such as sparse matrix computation [1] into the diffusion process is already a first step towards improving scalability.
> We will ensure this context and potential for future scaling optimizations are more prominently discussed in the revised manuscript.
>
> **W3: No bipartite representation baselines**
>
> To the best of our knowledge, no prior work on hypergraph generation using a diffusion model over bipartite graphs exists.
> We decided to focus directly on graph superposition because, already from a theoretical standpoint, it provides a more compact representation for hypergraphs.
> Indeed, Moon and Moser [4] show that the maximal number of maximal cliques in a graph with $N$ nodes is $\Theta(3^{\frac{N}{3}})$, which means that, in the most optimized case, we can store a hypergraph with $N$ nodes and $\approx 3^{\frac{3}{N}}$ hyperedges with $N^2$ numbers (to store the adjacency matrix).
> In contrast, a bipartite graph representation would have required storing the $N \cdot 3^\frac{N}{3}$ possible pairwise connections, which is far less effective.
> Indeed, an $N\times P$ matrix can represent a bipartite graph with $N$ nodes on one side and $P$ (representing hyperedges) nodes on the other. This representation is the biadjacency matrix. Unlike the adjacency matrix representation, only connexions between the two sides of the bipartite graph are represented in the biadjacency matrix.
>
> _References:_
>
> [1] Yiming Qin, Clément Vignac, Pascal Frossard: SparseDiff: Sparse Discrete Diffusion for Scalable Graph Generation. Trans. Mach. Learn. Res. 2025 (2025)
> [2] Clément Vignac, Igor Krawczuk, Antoine Siraudin, Bohan Wang, Volkan Cevher, Pascal Frossard: DiGress: Discrete Denoising diffusion for graph generation. ICLR 2023
> [3] Foggia, Pasquale. "An improved algorithm for matching large graphs." 3rd IAPR-TC15 workshop…. 2001.
> [4] Moon, John W., and Leo Moser. "On cliques in graphs." Israel journal of Mathematics 3.1 (1965): 23-28.
> [5] Murali Krishna Enduri: Navigating Social Networks: A Hypergraph Approach to Influence Optimization. COMPLEXIS 2024: 99-106
> [6] Feng, Song, et al. "Hypergraph models of biological networks to identify genes critical to pathogenic viral response." BMC bioinformatics 22.1 (2021): 287.
> [7] Lianghao Xia, Chao Huang, Chuxu Zhang: Self-Supervised Hypergraph Transformer for Recommender Systems. KDD 2022: 2100-2109

---

> > ### Author Rebuttal · Reviewer_fqY5 · 2026-04-04
> >
> > We appreciate the reviewer's responses and clarifications to the initial review. However, scalability remains a key concern limiting the impact of this work, and all of the questions raised in the initial review remain unanswered. I will maintain my score.

---

> > > ### Author Response · Authors · 2026-04-06
> > >
> > > We thank the reviewer for the acknowledgment.
> > > Below is our response to questions initially omitted due to space constraints:
> > >
> > > **Minor fixes:**
> > > We will cite [1, 2, 3] for exemplified hypergraph application domains and detailed application examples, and cite baseline state-of-the-art hypergraph generators (HYGENE, HyperPLR) in the introduction already.
> > >
> > > **Q1:**
> > > Clique expansion indeed causes information loss, as it indiscriminately expands all hyperedges into a single graph representation.
> > > For example, given two hyperedges over nodes (1, 2, 3) and (2, 3), clique expansion would be unable to recover the (2, 3) edge.
> > > In our superposition, we carefully choose one of multiple layers (graphs) into which to expand each hyperedge, so that the set of maximal cliques in each layer unambiguously gives the hyperedges assigned to it.
> > > For the example above, we split the hyperedges (1, 2, 3) and (2, 3) across different layers.
> > >
> > > **Q2:**
> > > Our method aims to restrain the overall number of layers while ensuring that creating the superposition of a hypergraph remains efficient.
> > > An optimal approach minimizing each graph's layer count would be dramatically more computationally expensive, requiring us to explore a combinatorial search space.
> > > Also, the number of layers for model training is given by the max number of layers across training hypergraphs, so despite a lower average number of layers, an optimal approach may still incur the same model training overhead as our greedy one.
> > > As noted in the paper, we require at most 6 layers for the datasets we tested.
> > > Figs. 9 and 10 show that our time to project a single graph into a superposition is generally under a second and requires only one attempt (no retries).
> > >
> > > **Q3:**
> > > Like mapping superpositions to hypergraphs, mapping hypergraphs to superpositions is lossless.
> > > We will update the highlighted paper sentence ("Given a graph-superposition, the corresponding hypergraph can be reconstructed with no loss of information [...]") to clarify that transforming a hypergraph to a superposition and back to a hypergraph is lossless as a whole.
> > > Based on layer counts and hyperedge assignments (Def. 3.4), each hypergraph has multiple valid superpositions that each map back to an isomorphism of the starting hypergraph.
> > >
> > > **Q4:**
> > > To identify generated hypergraphs isomorphic to the training ones, we use a process similar to the code for DiGress [4]: compare the degree sequences of the input graphs, and, if they match, perform a full isomorphism check using the vf2 algorithm [5], as implemented in the networkx Python library.
> > > We check isomorphism on the bipartite representations of the hypergraphs, which is equivalent to using the hypergraphs directly, but allows us to harness pre-existing algorithms.
> > > While checking graph isomorphism is in NP, the aforementioned approach is tractable for the tested datasets.
> > >
> > > **Q5:**
> > > Using maximal cliques does not create an inductive bias.
> > > Mapping a hyperedge to a clique is a mathematical equivalence that makes no assumptions about the dataset's natural topology.
> > > By definition, a hyperedge groups a set of nodes together, so an N-node clique in a standard graph losslessly encodes the relationship.
> > > Prior state-of-the-art methods also learn over decompositions of hypergraphs into normal graphs via clique expansion.
> > > However, standard clique expansion suffers from information loss when overlapping cliques merge.
> > > We solve the issue by projecting conflicting hyperedges into disjoint layers.
> > >
> > > **Q6:**
> > > Most parts of our graph transformer are permutation invariant.
> > > With Fig. 5 as a reference, we have:
> > > * **Input/output MLPs**: Invariant, as they apply point-wise and layer-wise.
> > > * **Graph-transformer** (main text $\theta_\text{single}$): Invariant, using identical weights across all superposition layers.
> > > * **XYTransformer** (main text $\theta_\text{multi}$): Inputs aggregate node/graph representations across layers, and given permutation invariant aggregations, the module and output embeddings are invariant.
> > > * **XxTransformer and YyTransformer** (main text $\theta_\text{XMix}$ and $\theta_\text{YMix}$): Not invariant; node/graph-level module inputs partly comprise sequenced layer-level representations.
> > >
> > > Since most model parameters belong to invariant modules, we believe the impact of the non-invariant section is limited, even if it remains a point of consideration for future work.
> > >
> > > _References:_
> > >
> > > [1] Enduri, Murali Krishna. "Navigating Social Networks: A Hypergraph Approach to Influence Optimization." COMPLEXIS. 2024.
> > >
> > > [2] Feng, Song, et al. "Hypergraph models of biological networks to identify genes critical to pathogenic viral response." BMC Bioinformatics 22.1
> > >
> > > [3] Xia, Lianghao, Chao Huang, and Chuxu Zhang. "Self-supervised hypergraph transformer for recommender systems." SIGKDD 2022
> > >
> > > [4] Vignac, Clement, et al. "DiGress: Discrete Denoising diffusion for graph generation." ICLR 2023
> > >
> > > [5] Foggia, Pasquale. "An improved algorithm for matching large graphs." 3rd IAPR-TC15 workshop

---

### Official Review · Reviewer_hJph · 2026-03-12

**Soundness:** 2
**Presentation:** 3
**Significance:** 2
**Originality:** 3
**Overall Recommendation:** 3
**Confidence:** 4

**Summary:**

The authors have introduced SuperHype, which is a pretty smart way to handle things. Instead of getting stuck in the usual traps, they’ve come up with a graph-superposition decomposition method. The idea is to represent these complex hypergraphs as multilayer graphs so they stay mathematically exact but are actually manageable for a diffusion model to generate. They have done the hypergraph synthesis by introducing a graph-superposition decomposition, a graph-superposition transformer, and hypergraph-specific enhancements such as clique-based auxiliary features and triplet aggregation. They tested it against six baselines across five different datasets. SuperHype is beating the older methods quite convincingly, especially when you look at graph and spectral metrics. It’s a solid step forward for anyone trying to generate complex relational data without the usual computational headache.

**Compliance With Llm Reviewing Policy:**

Affirmed.

**Final Justification:**

My main concerns regarding experiments and scalability remain only partially addressed. The current response provides useful context, but it does not add new empirical evidence on  real-world datasets, plus the authors have not demonstrated improved scalability beyond the current experimental setting. Without these additions, it is difficult to assess the practical utility and robustness of the work.                                                                              Therefore, while I appreciate the clarification and responsiveness, I consider my concerns only partially resolved overall and will keep my score unchanged.

**Key Questions For Authors:**

Refer weknesses

**Limitations:**

Yes

**Strengths And Weaknesses:**

**Strengths:**

1. The core concept is original. Instead of just doing the same old thing, the representation is done as a hypergraph as a disjoint union of maximal cliques across different layers. It’s a good way to bridge the gap between keeping the structure exact and making it easy for a graph-based model to actually generate it.

2. The paper hits the bottleneck in hypergraph generation. The authors explain clearly why the old bipartite or clique expansion methods are failing us. By positioning SuperHype as the solution to this tradeoff, they’ve made the paper very relevant for anyone working on higher-order data.

3. Empirical performance on synthetic benchmark like SBM, Tree, and Erdos-Renyi are looking quite promising. Especially when looked at the node-degree and spectral metrics.

**Weaknesses:**

1. Weak experimental evidence. The experiments are done mostly on synthetic datasets and the claims are thereafter unrealistic. Synthetic datasets are fine for a proof of concept, but the authors only tested one real-world dataset, ModelNet40 Bookshelf.
2. The scalability limitation is substantial. The paper acknowledges that the method works well mainly for hypergraphs with fewer than roughly 200–250 nodes and suffers from sharply increasing cost beyond that limit which is small to be considered as large data.
3. The projection condition is unclear. The condition for adding a hyperedge to a layer in Definition 3.4, $\mathcal{M}' = \mathcal{M} \cup {e}$, is the core of the projection algorithm. It requires that adding the edges of hyperedge $e$ creates exactly one new maximal clique ($e$ itself) and does not disturb any existing maximal cliques. This is a very restrictive condition, and its implications and the intuition behind it are not well-explained in the main text, making Algorithm 1 feel a bit like a black box.

---

> ### Author Rebuttal · Authors · 2026-03-31
>
> We thank the reviewer for their invaluable feedback and seek to address their concerns below.
>
> **W1: Weak experimental evidence**
>
> For our initial experiments, we chose to draw from the evaluation setting of our primary, state-of-the-art baseline, HYGENE.
> By adopting datasets used in their work (SBM, Ego, Hypertree, Erdos-Renyi, and the real-world ModelNet40 Bookshelf), we aimed to provide a direct, transparent comparison with the current standard in the field.
> Nevertheless, given the target of synthesizing hypergraphs with complete data patterns in a broad sense, we agree that demonstrating our model's capabilities across additional real-world datasets is of interest, and we will address this in future work.
>
> **W2: Scalability**
>
> Most graph synthesizers still face challenges in scaling to hypergraphs with many hundreds of nodes.
> State-of-the-art baselines like HYGENE face similar scalability challenges to our model and limit their evaluation to the same types of datasets.
> Despite performing diffusion over an exact representation (higher overhead, but better generation quality), which is not the case for HYGENE, SuperHype achieves comparable (often lower) training and sampling times, as shown by Table 8 in the appendix.
> While we view our framework as a foundational step for tractable hypergraph diffusion on exact representations, we certainly welcome further enhancements to push this scalability boundary.
> As highlighted in our reply to Reviewer m267's Q2, integrating techniques such as sparse matrix computation [1] into the diffusion process is already a first step towards improving scalability.
> We will ensure this context and potential for future scaling optimizations are more prominently discussed in the revised manuscript.
>
> **W3: Unclear projection condition**
>
> As correctly pointed out by the reviewer, the crux of the condition for adding a hyperedge to a specific layer (Definition 3.4) is that the hyperedge's nodes, when added, form a new maximal clique in that layer.
> The motivation behind the rule is that it allows unambiguous extraction of all hyperedges from a layer by finding the maximum cliques.
> Without this rule, finding all the hyperedges assigned to a layer would not be unambiguous. This rule, coupled with the fact that each hyperedge is integrated into exactly one layer, allows the starting hypergraph to be exactly reconstructed from the set of layer graphs.
> Moreover, while the rule is certainly restrictive in the context of an individual layer, using a small number of layers (up to 6 in our experiments) is sufficient to project most graphs on the first attempt (as shown in Figures 9 and 10).
> We will clarify this in the revised paper.
>
> _References:_
>
> [1] Yiming Qin, Clément Vignac, Pascal Frossard: SparseDiff: Sparse Discrete Diffusion for Scalable Graph Generation. Trans. Mach. Learn. Res. 2025 (2025)

---

> > ### Author Rebuttal · Reviewer_hJph · 2026-04-03
> >
> > Thank you for the rebuttal. The response helps clarify the intuition behind the projection condition, and I appreciate the explanation that the “new maximal clique” rule is necessary for unambiguous recovery of hyperedges from each layer.  This resolves part of my concern about Algorithm 1 feeling like a black box. My assessment by point:
> > 1.⁠ ⁠W1:  This is not really resolved. The authors’ reply is essentially, "We followed HYGENE’s evaluation setup for fair comparison, and broader real-world evaluation is future work." That explains why they chose the datasets, but it does not answer my core concern that the evidence is still mostly synthetic and only includes one real-world dataset.
> > 2.⁠ ⁠W2: This issue is only partially resolved. This point is only better contextualized by the author's argument that competing methods face similar limits and mention sparse methods as a future direction.
> > 3.⁠ ⁠W3: Meaningfully addressed. The author explains the intuition: the “new maximal clique” rule is needed so that hyperedges can be extracted unambiguously from each layer, and that exact reconstruction follows because each hyperedge is assigned to exactly one layer. The author also adds an empirical statement that a small number of layers is usually enough in practice. That does make the method less of a black box.
> > That said, my two main concerns remain largely unchanged. Overall, the rebuttal improves clarity, especially regarding the projection rule, but it does not sufficiently change my assessment of the paper’s empirical strength and scalability. Therefore, I am keeping my score unchanged.

---

> > > ### Author Response · Authors · 2026-04-06
> > >
> > > We wish to thank the reviewer for the thoughtful and detailed acknowledgment, and for engaging so constructively with our rebuttal.
> > > We are glad to hear that our explanation of the projection condition resolved concerns regarding W3 and that our discussion on scalability provided helpful context for W2.
> > > Finally, for W1, we want to reaffirm our commitment to exploring additional real-world datasets through future work or manuscript versions.

---

### Decision · Program_Chairs · 2026-04-30

**Decision:**

Accept (regular)

**Comment:**

The paper received four reviews, three positive and one negative (ratings: 3, 4, 5, 4). Based on the reviews, the paper has the following major merits:
1. The graph-superposition decomposition is an elegant, mathematically sound way to represent hypergraphs exactly (losslessly) as multi-layer graphs, avoiding the intractability of prior methods.
2. The proposed SuperHype consistently outperforms strong baselines in reproducing both local and global hypergraph connectivity patterns.

The paper has the following major weaknesses:
1. The method works well only for hypergraphs under ~200-250 nodes, limiting practical application to large real-world networks (a claimed use case).
2. Experiments rely heavily on synthetic data; only one real-world dataset is used, weakening claims of practical utility.
3. No comparison against lossless bipartite representations or latent-space diffusion models for hypergraphs. (The rebuttal has resolved the concern)

After carefully reading the paper and the four reviews, I believe that the idea is novel and the proposed method is effective in generating high-quality graphs. Reducing computational complexity could be a future work. Although the experiments include only one real dataset, there are three synthetic datasets that have been tested and shown the effectiveness of the proposed method.

In sum, I recommend accepting the paper.